# Chromosome evolution and the genetic basis of agronomically important traits in greater yam

Jessen V. Bredeson [1,18], Jessica B. Lyons [1,2,18], Ibukun O. Oniyinde[3], Nneka R. Okereke[4], Olufisayo Kolade[3], Ikenna Nnabue[4], Christian O. Nwadili[4], Eva Hřibová[5], Matthew Parker[6], Jeremiah Nwogha[4], Shengqiang Shu [7], Joseph Carlson[7], Robert Kariba[8,9], Samuel Muthemba [8,9], Katarzyna Knop[6], Geoffrey J. Barton [6], Anna V. Sherwood [6,16], Antonio Lopez-Montes[3,17], Robert Asiedu [3], Ramni Jamnadass[8,9], Alice Muchugi[8,9], David Goodstein [7], Chiedozie N. Egesi[3,4,10], Jonathan Featherston[11], Asrat Asfaw [3], Gordon G. Simpson[6,12], Jaroslav Doležel [5], Prasad S. Hendre [8,9], Allen Van Deynze [13], Pullikanti Lava Kumar[3], Jude E. Obidiegwu [4✉], Ranjana Bhattacharjee [3✉] & Daniel S. Rokhsar [1,2,7,14,15✉]

The nutrient-rich tubers of the greater yam, *Dioscorea alata* L., provide food and income security for millions of people around the world. Despite its global importance, however, greater yam remains an orphan crop. Here, we address this resource gap by presenting a highly contiguous chromosome-scale genome assembly of *D. alata* combined with a dense genetic map derived from African breeding populations. The genome sequence reveals an ancient allotetraploidization in the *Dioscorea* lineage, followed by extensive genome-wide reorganization. Using the genomic tools, we find quantitative trait loci for resistance to anthracnose, a damaging fungal pathogen of yam, and several tuber quality traits. Genomic analysis of breeding lines reveals both extensive inbreeding as well as regions of extensive heterozygosity that may represent interspecific introgression during domestication. These tools and insights will enable yam breeders to unlock the potential of this staple crop and take full advantage of its adaptability to varied environments.

[1] Department of Molecular & Cell Biology, University of California, Berkeley, CA 94720, USA. [2] Innovative Genomics Institute, Berkeley, CA, USA. [3] International Institute of Tropical Agriculture, PMB 5320, Oyo Road, Ibadan, Nigeria. [4] National Root Crops Research Institute (NRCRI), Umudike, Nigeria. [5] Institute of Experimental Botany of the Czech Academy of Sciences, Centre of the Region Haná for Biotechnological and Agricultural Research, Šlechtitelů 31, CZ-77900 Olomouc, Czech Republic. [6] School of Life Sciences, University of Dundee, Dundee, UK. [7] DOE Joint Genome Institute, Berkeley, CA, USA. [8] World Agroforestry (CIFOR-ICRAF), Nairobi, Kenya. [9] African Orphan Crops Consortium, Nairobi, Kenya. [10] Cornell University, Ithaca, NY 14850, USA. [11] Agricultural Research Council, Biotechnology Platform, Pretoria, South Africa. [12] James Hutton Institute, Dundee, UK. [13] University of California, Davis, Davis, CA 95616, USA. [14] Okinawa Institute of Science and Technology, Onna, Okinawa, Japan. [15] Chan-Zuckerberg BioHub, 499 Illinois St., San Francisco, CA 94158, USA. [16] Present address: Department of Biology, University of Copenhagen, Copenhagen, Denmark. [17] Present address: International Trade Center, Accra, Ghana. [18] These authors contributed equally: Jessen V. Bredeson, Jessica B. Lyons. ✉email: ejikeobi@yahoo.com; r.bhattacharjee@cgiar.org; dsrokhsar@gmail.com

Yams (genus *Dioscorea*) are an important source of food and income in tropical and subtropical regions of Africa, Asia, the Pacific, and Latin America, contributing more than 200 dietary calories per capita daily for around 300 million people[1]. Yam tubers are rich in carbohydrates, contain protein and vitamin C, and are storable for months after harvesting, so they are available year-round[2,3]. World annual production of yam in 2018 was estimated at 72.6 million tons (FAOSTAT 2020). Over 90% of global yam production comes from the 'yam belt' (Nigeria, Benin, Ghana, Togo, and Cote d'Ivoire) in West Africa, where yam's importance is demonstrated by its vital role in traditional culture, rituals, and religion[3–5]. While yams are primarily dioecious, and hence obligate outcrossers, they are vegetatively propagated, allowing genotypes with desirable qualities (disease resistance, cooking quality, nutritional value) to be maintained over subsequent planting seasons.

Greater yam (*Dioscorea alata* L.), also called water yam, winged yam, or ube, among other names, is the species with the broadest global distribution[1]. *D. alata* is thought to have originated in Southeast Asia and/or Melanesia[2,6]. It was introduced to East Africa as many as 2000 years ago and reached West Africa by the 1500s[2,7]. Several traits of greater yam make it particularly valuable for economic production and an excellent candidate for systematic improvement. It is adapted to tropical and temperate climates, has a relatively high tolerance to limited-water environments, and no other yam comes close for yield in terms of tuber weight. Greater yam is easily propagated, its early vigor prevents weeds, and its tubers have high storability[8]. The tubers of *D. alata* possess high nutritional content relative to other *Dioscorea* spp[9,10].

Over the last two decades, global yam production has doubled, but these increases have predominantly been achieved through the expansion of cultivated areas rather than increased productivity[1] (FAOSTAT 2020). To meet the demands of an ever-growing population and tackle the threats that constrain yam production, the rapid development of improved yam varieties is urgently needed[11]. Conventional breeding for desired traits in greater yam is arduous, however, due to its long growth cycle and erratic flowering, and is further complicated by the polyploidy common in this species[12–14]. Efforts are currently underway by breeders to develop greater yam varieties with improved yield, resistance to pests and diseases, and tuber quality consistent with organoleptic preferences such as taste, color, and texture[11]. A critical challenge for greater yam is its high susceptibility to the foliar disease anthracnose, caused by the fungal pathogen *Colletotrichum gloeosporioides* Penz. Anthracnose disease is characterized by leaf necrosis and shoot dieback, and can cause losses of over 80% of production[15–18]. Anthracnose disease affects greater yam more than other domesticated yams; moderate resistance to this disease is present, however, in greater yam landraces and breeder's lines[19,20]. High-quality genomic resources and tools can facilitate rapid breeding methods for greater yam improvement with huge potential to impact food and nutritional security, particularly in Africa.

Here, we describe a chromosome-scale reference genome sequence for *D. alata* and a dense 10k marker composite genetic linkage map from five populations involving seven distinct parental genotypes. Comparison of the *D. alata* reference genome sequence with the recently sequenced genomes of the distantly related *D. rotundata*[21] and *D. zingiberensis*[22] reveals substantial conservation of chromosome structure between *D. alata* and *D. rotundata*, but considerable rearrangement relative to the more deeply divergent *D. zingiberensis* lineage. Analysis of the *D. alata* genome sequence supports the existence of ancient polyploidy events shared across Dioscoreales. Using a non-parametric statistical test for biased gene loss between subgenomes, we infer that all *Dioscorea* share an ancient paleo-allotetraploidy, which was followed by species-specific chromosome rearrangements. We use genomic and genetic resources to identify nine QTL for anthracnose resistance and tuber quality traits. Our dense multi-parental genetic map complements the maps previously used for QTL mapping for anthracnose resistance[23–25] and sex determination[26]. These tools and resources will empower breeders to use modern genetic tools and methods to breed the crop more efficiently, thereby accelerating the release of improved varieties to farmers.

## Results and discussion

**Genome sequence and structure**. We generated a high-quality reference genome sequence for *D. alata* by assembling whole-genome shotgun sequence data from PacBio single-molecule continuous long reads (234× coverage in reads with 15.1 kb N50 read length), with short-read sequencing for polishing and additional mate-pair linkage (see Methods, Table 1, Supplementary Note 1, Supplementary Data 1). High-throughput chromatin conformation contact (HiC) data and a composite meiotic linkage map (see below) were used to organize the contigs (N50 length 4.5 Mb) into $n = 20$ chromosome-scale sequences, matching the observed karyotype, with each pair of homologous chromosomes represented by a single haplotype-mosaic sequence (Supplementary Figs. 1–3). The genome assembly spans a total of 479.5 Mb, consistent with estimates of 455 ± 39 Mb by flow cytometry[13], and 477 Mb by *k*-mer-based analyses (Table 1, Supplementary Note 1). The chromosome-scale 'version 2' assembly is available via Yam-Base (ftp://yambase.org/genomes/Dioscorea_alata) and Phytozome (https://phytozome-next.jgi.doe.gov/info/Dalata_v2_1), replacing the early 'version 1' draft released in those databases in 2019.

The genomic reference genotype, TDa95/00328, is a breeding line from the Yam Breeding Unit of the International Institute of Tropical Agriculture (IITA), Ibadan, Nigeria. It is moderately resistant to anthracnose[23,27] and has been used as a parent

### Table 1 Assembly and annotation statistics.

| Assembly statistic | Value |
|---|---|
| Scaffold sequence total/count | 480.0 Mb/25 |
| Scaffold N50 length/count | 24.0 Mb/9 |
| Scaffold N90 length/count | 19.5 Mb/18 |
| Contig sequence total/count | 479.5 Mb/532 |
| Contig N50 length/count | 4.5 Mb/31 |
| Contig N90 length/count | 565.0 kb/126 |

| Annotation statistic | Value |
|---|---|
| Primary transcripts[a] (loci) | 25,189 |
| Alternate transcripts[b] | 13,414 |
| Total transcripts | 3860 |
| *Primary transcripts* | |
| Average number of exons | 5.5 |
| Median exon length (bp) | 156 |
| Median intron length (bp) | 151 |
| Number of complete genes | 24,614 |
| Number of incomplete genes with start codon | 218 |
| Number of incomplete genes with stop codon | 281 |
| *Gene model support* | |
| Number of genes with Pfam annotation | 19,599 |
| Number of genes with Panther annotation | 23,183 |
| Number of genes with KOG annotation | 10,939 |
| Number of genes with KEGG Orthology annotation | 6849 |
| Number of genes with E.C. number annotation | 7654 |

[a]The longest transcript for each protein-coding gene.
[b]All other splice isoforms.

frequently in crossing programs. TDa95/00328 is diploid with $2n = 2x = 40$, as confirmed by chromosome counting (Supplementary Fig. 2) and genetically by segregation of AFLP[23]. The reference accession exhibits long runs of homozygosity due to recent inbreeding (Supplementary Fig. 4); outside of these segments we observe 7.9 heterozygous sites per kilobase.

To corroborate our genome assembly and provide tools for genetic analysis, we generated ten genetic linkage maps from eleven mapping populations that involved seven distinct parents segregating for relevant phenotypic traits (one of the maps combined two small, related mapping populations; Table 2, Supplementary Tables 1 and 2; see below). These mapping populations were generated from biparental crosses performed at IITA, with 32–317 progeny per cross. Genotyping was performed using sequence tags generated with DArTseq (Diversity Arrays Technology Pty), mapped to the genome assembly, and filtered (Methods, Supplementary Note 2), producing 13,584 biallelic markers that segregate in at least one of our mapping populations (Supplementary Table 3).

The 20 linkage groups derived from individual maps corroborated the sequence-based genome assembly and were particularly useful for interpreting HiC linkage between chromosome arms and determining their correct intrachromosomal orientations. These features were difficult to organize using HiC alone, due to strong 'Rabl' configurations (Fig. 1a, and Supplementary Figs. 1 and 5)—the three-dimensional chromatin structure characterized by polarized centromere or telomere clustering on the inner membranes of cell nuclei[28–30]—that led to contacts between the distal regions of chromosome arms (see below). The ten genetic maps were highly concordant (Fig. 1b; Kendall's tau correlation coefficients = 0.9091–0.9626), and we combined them into a single composite linkage map using five maps that capture the genetic diversity of the seven distinct parents (Supplementary Table 3). The composite map spans 1817.9 centimorgans, accounting for a total of 2178 meioses (1089 individuals), and includes 10,448 well-ordered (Kendall's tau = 0.9989; Supplementary Fig. 6) markers (excluding markers genotyped in individual crosses that were discordant post-imputation and/or were not phaseable) (Methods, Supplementary Note 2). This is the highest resolution genetic linkage map for *D. alata* produced to date.

The *D. alata* reference genome sequence encodes an estimated 25,189 protein-coding genes, based on an annotation that took advantage of both existing and the *D. alata* transcriptome resources generated in this study as well interspecific sequence homology (Table 1, Methods, Supplementary Note 3). With a benchmark set of embryophyte genes[31,32], we estimate that the *D. alata* gene set is 97.8% complete, with 1.5% gene fragmentation. While BUSCO methodology suggests that only 0.7% of the genes are missing, this is an overestimate, since some of these nominally-missing genes are detected by more sensitive searches (Supplementary Note 3). Our transcriptome datasets include short-read RNA-seq as well as 626,000 long, single-molecule direct-RNA sequences from twelve TDa95/00328 tissues. The transcriptome data identified 13,414 alternative transcripts. The great majority of genes have functional assignments through Pfam ($n = 19,599$) and Panther ($n = 23,183$) (Table 1).

Within chromosomes, protein-coding gene and transposable element densities are strongly anticorrelated (Pearson's $r = -0.885$), with gene loci concentrated in the highly-recombinogenic distal chromosome ends (Pearson's $r = +0.823$) and transposable elements, particularly Ty3/metaviridae and Ty1/pseudoviridae LTRs and other unclassified repeats, are enriched in the recombination-poor pericentromeres (Pearson's $r = -0.718$) (Fig. 1c, Supplementary Fig. 6, Supplementary Table 4). Homopolymers and simple-sequence repeats, however, were positively correlated with gene

**Table 2 Mapping populations used in this study.**

| Pop. ID[a] | Inst. | Seed parent | Pollen parent | Putative parental relation[b] | Trait(s) studied |
|---|---|---|---|---|---|
| TDa1401 | IITA | TDa05/00015 | TDa99/00048 | Half avuncular | Anthracnose susceptibility (field, DLA) |
| TDa1402 | IITA | TDa05/00015 | TDa02/00012 | Fourth-degree relative | Anthracnose susceptibility (field[c], DLA), tuber fresh weight, tuber dry weight, tuber flesh color, tuber oxidation, dry matter content |
| TDa1403 | IITA | TDa00/00005 | TDa02/00012 | Third-degree relative | Anthracnose susceptibility (field, DLA), tuber fresh weight, tuber dry weight, tuber flesh color, tuber oxidation, dry matter content |
| TDa1419 | IITA | TDa99/00240 | TDa02/00012 | Unrelated | Anthracnose susceptibility (field, DLA[c]), tuber fresh weight, tuber dry weight, tuber flesh color, tuber oxidation[c], tuber flesh color |
| TDa1427 | IITA | TDa95/00328 | TDa02/00012 | Unrelated | Anthracnose susceptibility (field, DLA), tuber fresh weight, tuber flesh color, tuber oxidation, dry matter content |
| TDa1401B | NRCRI | TDa05/00015 | TDa99/00048 | Half avuncular | Anthracnose susceptibility (DLA), presence of corm, ability of corm to separate, corm type, tuber shape, tuber size[c], tuber surface texture, roots on tuber, placement of roots on tuber |
| TDa1506 TDa1621 | NRCRI NRCRI | TDa05/00015 | TDa02/00012 | Fourth-degree relative | (In TDa1506) Anthracnose susceptibility (DLA), presence of corm, ability of corm to separate, corm type, tuber shape, tuber size, tuber surface texture, roots on tuber, placement of roots on tuber |
| TDa1512 TDa1603 | NRCRI NRCRI | TDa00/00005 | TDa01/00039 | Parent– offspring | (In TDa1512) Anthracnose susceptibility (DLA), presence of corm, ability of corm to separate, corm type, tuber shape, tuber size, tuber surface texture, roots on tuber, placement of roots on tuber |
| TDa1610 | NRCRI | TDa99/00240 | TDa02/00012 | Unrelated | – |

*Pop. ID* mapping population identifier, *Inst.* institution that grew the plants and performed the phenotyping, *Parental Relation* parental relatedness as assessed in this study, *DLA* detached leaf assay.
[a]The first two digits in a population ID denote the year of crossing. All crosses were performed at IITA and, where applicable, progeny were sent to NRCRI as botanical seeds. For mapping populations that share parents across institutes, subsets of the progeny were sent to NRCRI. For NRCRI crosses with the same parents but different population IDs (TDa1506/1621 and TDa1512/1603), the second population ID was assigned to those individuals from a cross performed with the same parents in a subsequent year. We treated these pairs as single populations for the purposes of linkage mapping, but individually for QTL analyses.
[b]Putative parental relations derived from Fig. 4.
[c]Traits for which significant QTL were identified (see Table 3).

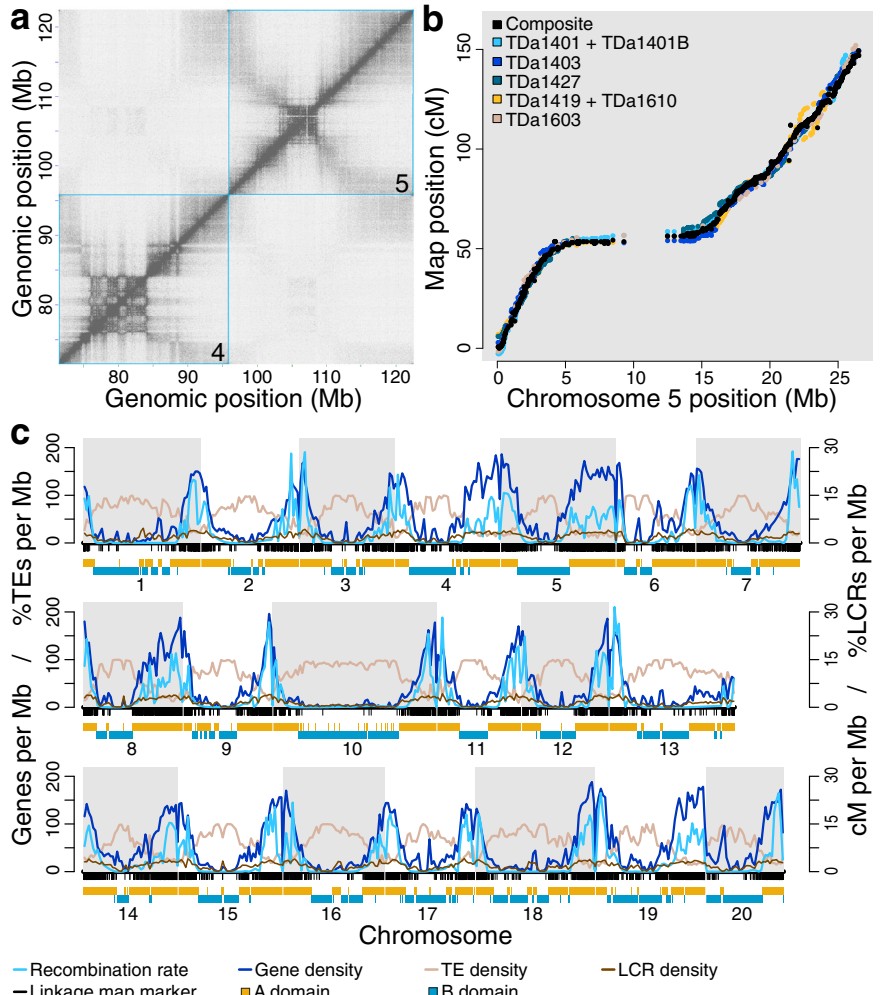

**Fig. 1 *D. alata* genome structure and recombination. a** HiC contact matrix of TDa95/00328 chromosomes 4 and 5. Within chromosomes, the band of high contact density along the diagonal reflects the well-ordered underlying assembly. The checkerboard pattern observed between 75 and 85 Mb indicates chromatin domain A/B compartmentalization[156] within chromosome 4. The winged pattern observed within chromosomes, particularly chromosome 5, showing elevated contact densities between chromosome ends is typical of Rabl-structured chromosomes in the nucleus[29]. Chromosomes are outlined with cyan boxes. Each pixel represents the intersection between a pair of 50 kb loci along the chromosomes. The density of contacts between two loci is proportional to pixel color, with darker pixels representing more contacts and lighter representing fewer. **b** A composite genetic linkage map (black points), integrating five mapping populations (colored points, legend), is shown for chromosome 5. The maps exhibit highly concordant marker orders (Kendall's tau correlations between 0.9091 and 0.9626) and validate the large-scale correctness of the chromosome-scale assembly. The sigmoidal shape of the maps along the physical chromosome reflects suppressed recombination within the pericentromere. Individual component maps were scaled and shifted vertically to display their marker-order concordance. **c** The *D. alata* chromosome landscape is shown. Transposable elements (TEs; tan lines, left *Y*-axis) are enriched within the pericentromeres; while low-complexity repeat (LCR; brown, right *Y*-axis), protein-coding gene (dark blue line, left *Y*-axis), and meiotic recombination (cyan lines, right *Y*-axis) densities are elevated nearer the chromosome ends. Densities were computed using 500 kb bins. Composite map marker positions are shown as black ticks under the *X*-axis, with A/B chromatin compartment structure drawn below (A compartment domains in gold; B domains in dark cyan). cM centiMorgan, Mb megabase. Source data are provided as a Source Data file.

(Pearson's $r = +0.838$) and recombination (Pearson's $r = +0.728$) densities.

Analysis of chromatin conformation capture (HiC) data reveals the structure of interphase chromosomes in *D. alata* (Methods, Supplementary Note 4). We find that all chromosomes adopt a Rabl-like configuration (Supplementary Fig. 5) in which each chromosome appears 'folded' in the vicinity of the centromere, as (1) chromatin contacts are enriched among chromosome ends and (2) these chromosome ends are depleted of contacts with the pericentromeres (see also refs. 28–30). *D. alata* chromosomes also show alternating A/B chromatin compartmentalization, as is demonstrated in several other plant species[33]. In *D. alata*, the gene-rich distal regions of each chromosome are generally spanned by open A domains (between gene density and A/B

domain status, Pearson's $r = +0.686$), while the relatively gene-poor and transposon-rich pericentromeres are characterized by closed B domains that are often punctuated by smaller A domains (Supplementary Fig. 7).

**Comparative analysis and paleopolyploidy.** Comparison of the *D. alata* genome sequence and protein-coding annotation with those of white yam (*D. rotundata*[21], also known as Guinea yam), bitter yam (*D. dumetorum*[34]), and peltate yam (*D. zingiberensis*[22]) highlights the completeness of our sequence and annotation and the extensive sequence divergence across the genus. Among the *Dioscorea* species sequenced to date, the annotation of *D. alata* appears to be the most complete (Supplementary Table 5,

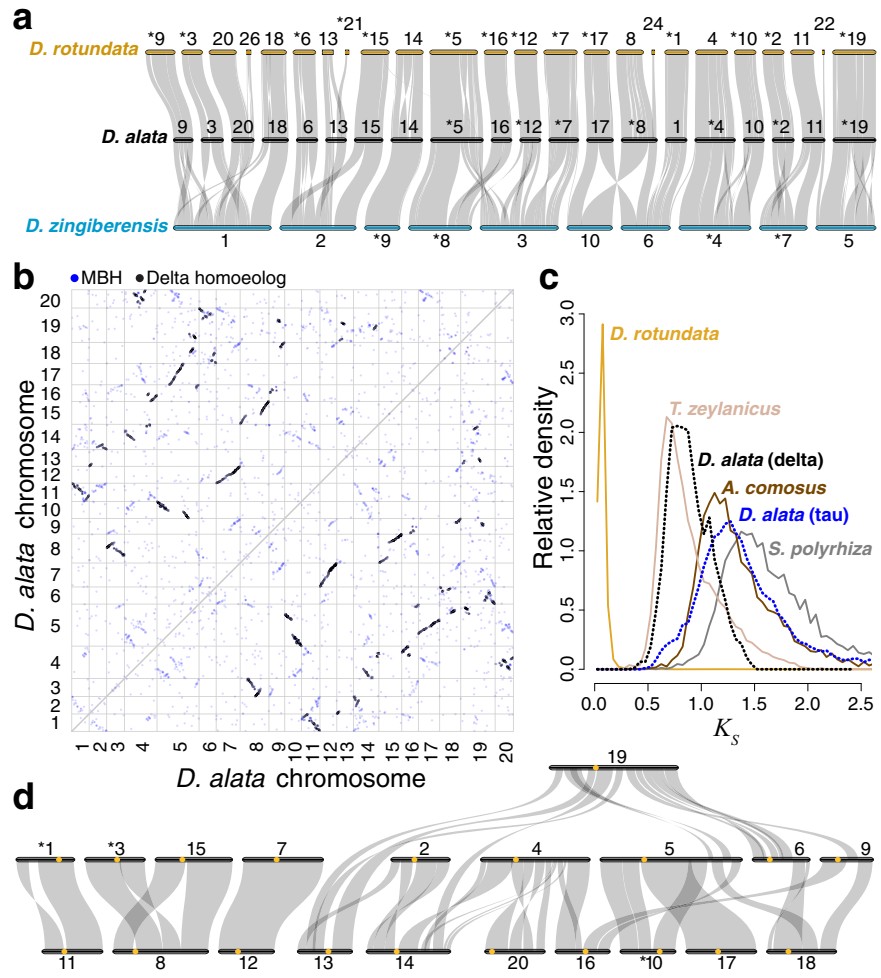

**Fig. 2 Dioscoreaceae chromosome evolution. a** Ribbon diagram demonstrating conserved chromosomal synteny and large-scale segmental collinearity (semi-transparent gray ribbons) between *Dioscorea alata* (black horizontal bars), *D. rotundata* (gold), and *D. zingiberensis* (cyan) one-to-one orthologous gene pairs. Only *D. rotundata* sequences with five or more collinear genes are shown. To improve visual clarity, some chromosomes, marked with asterisks, were reverse complemented with respect to their assembled sequences. Chromosome sizes are proportional to the number of annotated genes. **b** Dot plot showing evidence of two whole-genome duplications exposed by TDa95/00328 intragenomic comparison. Each point represents a mutual best-hit (MBH) gene pair and each white box (outlined in grey) represents the intersection of two chromosomes. Homoeology from the recent Dioscoreaceae delta duplication is shown in black and the ancient, core monocot tau duplication can be seen as clusters in blue (see also, Supplementary Fig. 9). **c** The synonymous substitution rate ($K_S$) histograms for orthologous (solid lines) or homoeologous (dotted lines) gene pairs between *D. alata* and select species comparators are shown: *D. rotundata* ($n = 14{,}889$), *T. zeylanicus* ($n = 9013$), *D. alata* delta ($n = 1578$), *A. comosus* ($n = 6405$), *D. alata* tau ($n = 404$), and *S. polyrhiza* ($n = 4973$). The *D. alata–D. rotundata* ortholog density was rescaled by 0.25 to emphasize other comparisons. **d** Shared segmental homoeology (semi-transparent gray) between *D. alata* chromosomes (black horizontal bars) resulting from the delta duplication is depicted with a ribbon diagram, as in panel **a**, but with putative centromere positions now included as gold circles (Supplementary Data 2). Source data are provided as a Source Data file.

Supplementary Note 3). For example, *D. alata* has the fewest missing conserved gene families in cross-species comparisons within Dioscoreaceae (53 in *D. alata* compared with 385 for *D. zingiberensis* and 595 for *D. rotundata*) and in cross-monocot comparisons (7 in *D. alata* compared with 99 in *D. zingiberensis* and 110 in *D. rotundata*) (Supplementary Fig. 8). These metrics combine genome assembly completeness and accuracy with exon-intron structure predictions based, in part, on transcriptome resources.

At the nucleotide level, *D. alata* coding sequences exhibit 97.4%, 93.6%, and 86.5% identity with *D. rotundata*, *D. dumetorum*, and *D. zingiberensis*, corresponding to median synonymous substitution ($K_S$) rates of 0.064, 0.163, and 0.389, respectively. These measures are consistent with *D. zingiberensis* being a deeply branching outgroup to the clade formed by *D. alata*, *D. rotundata*, and *D. dumetorum* (see also Supplementary Table 6), and highlights the ~60 My old divergences within the genus *Dioscorea*.

The medicinal plant *Trichopus zeylanicus* (common name 'Arogyappacha' in India, meaning 'the green that gives strength')[35] is a more distantly related member of the Dioscoreaceae family, with 77.9% identity and median $K_S$ of 0.804.

The ($n = 20$) chromosome sequences of *D. alata* and *D. rotundata*[21,36,37] are in 1:1 correspondence, and are highly collinear (Fig. 2a, Supplementary Fig. 9a). The few intra-chromosome differences observed could represent bona fide rearrangements between species or, possibly, imperfections in the *D. rotundata* v2 assembly[21] that could have arisen from the reliance on linkage mapping to order and orient *D. rotundata* scaffolds, especially in recombination-poor pericentromeric regions of the genome. Under the assumption that *D. rotundata* chromosomes are in 1:1 correspondence with *D. alata* chromosomes, we can provisionally assign four large but unmapped *D. rotundata* scaffolds to chromosomes (Fig. 2a). We found one inter-chromosome difference (not present in the *D. rotundata* v1

assembly[37]), which requires further study (Supplementary Fig. 9a). While the draft *D. dumetorum* genome assembly is not organized into chromosomes, comparison with the *D. alata* reference sequence shows that the two genomes are locally collinear on the scale of the *D. dumetorum* contigs, with only one discordance (Supplementary Fig. 9b). This observation suggests a provisional organization of the *D. dumetorum* contigs into probable chromosomes. Notably, the distantly related *D. zingiberensis* has a haploid complement of $n = 10$ (ref. [38]), compared with $n = 20$ found in *D. alata*, *D. rotundata*[21,39], and *D. dumetorum*[2,40]. We find that the *D. zingiberensis* chromosomes[22] were formed from ancestral, *D. alata*-like chromosomes and/or chromosome arms by combinations of end-to-end and centric fusions and translocations (Fig. 2a, Supplementary Fig. 9c).

We found evidence for two ancient paleotetraploidies in the *D. alata* lineage. These duplications evidently preceded the origin of the genus, since all *Dioscorea* genome sequences show one-to-one orthology (Supplementary Fig. 9a–c, Supplementary Note 5). The most recent paleotetraploidy is apparent from extensive collinear paralogy in *D. alata* (Fig. 2b) and coincides with the genome duplication recently described in *D. zingiberensis*[22] and previously identified based on transcriptome analysis of *D. villosa* in the context of one thousand plant transcriptomes as DIV1-alpha[41], but not found in an earlier analysis that included the *D. opposita* transcriptome[42]. Following the common use of Greek letters to denote plant polyploidies, we designate this *Dioscorea* lineage duplication as 'delta.' The median sequence divergence between 1,578 delta paralogs in *D. alata* is $K_S = 0.869$ substitutions/site (Fig. 2c). While comparisons with the draft genome assembly of *T. zeylanicus* ($K_S = 0.804$ to *D. alata*) further suggest that the delta paleotetraploidy may have preceded the origin of the family Dioscoreaceae, the fragmentation of the *T. zeylanicus* assembly precludes a definitive assessment. The timing of the delta duplication (estimated to be 64 Mya[22]) is contemporaneous with the K/T boundary and a cluster of other successful paleopolyploidies[43].

Analysis of the *D. alata* genome sequence reveals large-scale genomic reorganization after the delta duplication. *D. alata* chromosomes preserve long collinear paralogous segments arising from the delta paleotetraploidy event, and the genomic organization of these segments reveals large-scale rearrangements after whole-genome duplication (Fig. 2d, Supplementary Data 2). These include cases of one-to-one whole-chromosome paralogs, (chromosomes 1 and 11; 7 and 12) as well as examples of centric insertion (e.g., the paralog of chromosome 3 was inserted within the paralog of chromosome 15 to form chromosome 8; the paralog of chromosome 17 was inserted into the paralog of chromosome 10 to form most of the chromosome 5). Other large-scale rearrangements are evident, including apparent end-to-end 'fusions' (or more properly translocations[44]). Taken together, these paralogies provide further evidence for the delta duplication.

Genome duplication can occur by two distinct evolutionary mechanisms[45]: allotetraploidy (genome duplication after hybridization of two distinct diploid progenitors) or auto-tetraploidy (genome duplication within a single species). Since hybridization brings together genomes with distinct epigenetic properties[46], a hallmark of ancient allotetraploidy is differential evolution of the homoeologous chromosome sets ('subgenomes') inherited from distinct progenitor species. In particular, paleo-allotetraploids may exhibit asymmetric gene loss (or conversely, gene retention) between subgenomes, often referred to as 'biased fractionation'[45,47,48]. While the observation of asymmetric gene retention is considered positive evidence for paleo-allotetraploidy[45], a lack of detectable asymmetry in gene loss can be consistent either with autotetraploidy or with

allotetraploidy that is recent and/or involved hybridization of closely related progenitors species.

To test for patterns of differential gene retention that are diagnostic of paleo-allotetraploidy, we analyzed 15 robust pairs of paralogous *D. alata* segments (each with more than 40 paralogous genes) from the delta duplication, drawn from 11 distinct chromosome pairs. We observe a bimodal distribution of retention rates across these 30 chromosomal segments relative to the inferred unduplicated gene complement (Methods, Supplementary Note 5), with peaks at 0.63 and 0.48 (Supplementary Fig. 10). Importantly, for each of the 11 homoeologous chromosome pairs, one paralog has a high retention rate and the other low (Supplementary Table 7). Such a paired distribution of high and low-retention chromosomes is unexpected under the null (autotetraploid) model of uncorrelated gene loss ($p = 2.9 \times 10^{-3}$; $k = 11$, $n = 11$) (Supplementary Table 8, Supplementary Note 5). Analysis of the other *Dioscorea* genomes yields consistent results (Supplementary Tables 7 and 8).

Our finding of consistent patterns of differential gene retention between homoeologous chromosomes (1) allows us to reject the autotetraploid hypothesis, and (2) provides positive support for a paleo-allotetraploid scenario for the ancient delta genome duplication in *Dioscorea*. Under this paleo-allotetraploid scenario, the high- and low-retention chromosomes of *Dioscorea* spp. represent the descendants of the ancestral chromosomes of the two progenitors (now subgenomes). Since our method does not require an extant relative of the unduplicated progenitors[49] it can be applied to other ancient genome duplications, with the caveat that not all allotetraploidizations may trigger asymmetric gene loss[48,50].

In addition to delta, the *D. alata* genome sequence also displays relicts of a more ancient genome-wide duplication in the form of nearly-collinear ancient paralogous segments with median $K_S = 1.21$ substitutions per site (Fig. 2b, c). We identify this duplication with the famed 'tau' duplication shared by other core monocots, including grasses[50], pineapple (*Ananas comosus*[51]), oil palm (*Elaeis guineensis*[52]), and asparagus (*Asparagus officinalis*[53]) but not duckweed (*Spirodela polyrhiza*[54]). The tau duplication has also been noted in transcriptome analyses[41,42]. The clear 2:2 pattern of orthology between yam, pineapple, and oil palm (Supplementary Fig. 9d, e) confirms that these three lineages have each experienced one lineage-specific whole-genome duplication (delta, sigma, and p, respectively) since they diverged from each other. This pattern implies that relicts of any earlier duplications observed in these species must represent shared events. Since Dioscoreales is one of several early-branching core monocot lineages (only Petrosaviales branches earlier), the discovery of tau in yam implies that this duplication likely preceded the divergence of the core monocot clade (Supplementary Figs. 9f and 11). (Since tau occurred close in time to the divergence of the non-Petrosaviales core monocots, the combination of tau and the respective lineage-specific duplications produce 4:4 patterns of paralogy in dot plots. See Supplementary Fig. 9d, e)

**QTL mapping.** To demonstrate the utility of our dense linkage maps and high-quality *D. alata* reference genome sequence for advancing greater yam breeding, we searched for quantitative trait loci (QTL) for resistance to anthracnose disease and several tuber quality traits (dry matter, oxidation, tuber color, corm type, and other traits). Our mapping populations were generated in controlled crosses by yam breeders at IITA, Nigeria, using parents from the yam breeding program (Table 2, Supplementary Table 1). Phenotyping was performed in Nigeria at IITA Ibadan and NRCRI in Umudike (Methods, Supplementary Note 6). Leveraging the ability to clonally propagate individuals, we

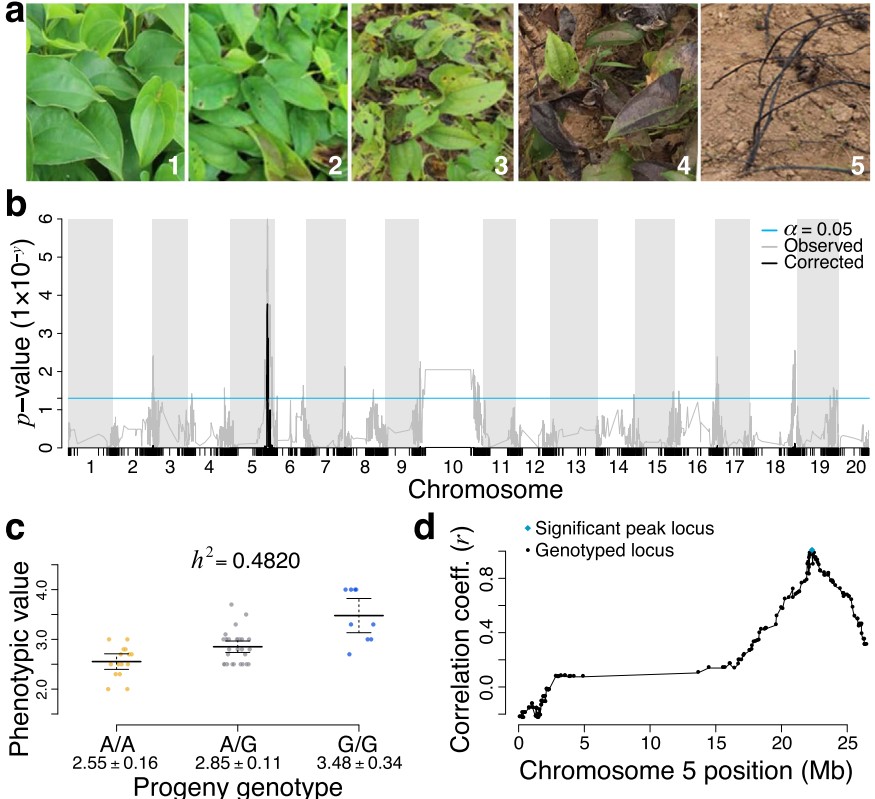

**Fig. 3 Quantitative trait locus for anthracnose resistance. a** Exemplars of the yam anthracnose disease (YAD) field assessment severity rating scale (scored 1–5) used at IITA in Ibadan, Nigeria. **b** Genome-wide QTL association scan for YAD resistance in the TDa1402 genetic population ($n = 53$ biologically independent samples) for the year 2017. A statistically significant association (corrected $p = 1.69 \times 10^{-4}$) was found on chromosome 5, at 23.3 Mb. Per-locus Wald statistic-based logistic regression significance values (gray line) were corrected for multiple testing (black line) via max($T$) adjustment with $1 \times 10^{6}$ permutations. The minimum significance threshold ($\alpha = 0.05$) is represented with a cyan horizontal line. **c** Effect plot for the peak locus on chromosome 5 at 23.3 Mb, the genotypes ($X$-axis) of which explain 48.2% of the observed phenotypic variance (i.e., narrow-sense heritability, $h^2$), suggests that an increased dose of the 'A' allele is associated with lower severity of YAD. Centerline and whisker plots, and their corresponding statistics ($X$-axis), represent the mean ± 95% confidence intervals. **d** Plot showing the strength of linkage disequilibrium (LD) between the peak marker (cyan diamond) and other loci (black points) in chromosome 5. LD was calculated as Pearson's correlation ($r$) between alleles. Source data are provided as a Source Data file.

measured multiple traits over the years 2016–2019. Our QTL analyses exploited the imputed genotypes derived from our dense linkage maps. In total, we found eight distinct QTL: three for anthracnose resistance and five for tuber traits (Fig. 3, Table 3, Supplementary Figs. 12–13).

**QTL for anthracnose resistance**. Yam Anthracnose Disease (YAD), or yam dieback, is a major disease afflicting yams caused by the fungus *Colletotrichum gloeosporioides*[15,18]. Greater yam is particularly susceptible to YAD, although resistance has been shown to vary among *D. alata* genotypes[55]. We sought QTL for YAD resistance using field trials in five mapping populations and detached leaf assays in eight mapping populations (Table 2, Methods, Supplementary Note 6). While most of these populations did not show significant QTL, we found three significant anthracnose resistance QTL in two of them.

In field trials of the TDa1402 population, we found a major QTL on chromosome 5 ($p = 1.69 \times 10^{-4}$) that explains 48.2% of phenotypic variance in the 2017 data, with an additive effect (Fig. 3a–c), and a minor QTL on chromosome 19 (Supplementary Fig. 12a–c) that explains 29.9% of the variance in the 2018 data ($p = 1.25 \times 10^{-2}$). Although anthracnose response and resistance are poorly understood in yams, studies in other species suggest potential candidate genes overlapping these QTL intervals, including a gene (Dioal.05G183500) on chromosome 5 that

encodes a receptor-like EIX1/2 protein, which is a member of the LRR (leucine-rich-repeat) superfamily of plant disease resistance proteins[56], and genes on chromosome 19 that encode members of the EMSY-LIKE family of immune regulators of fungal disease resistance[57,58] (Dioal.19G063700), three NB-ARC domain-containing *R*-gene analog (RGA) disease resistance protein-encoding genes[59] (Dioal.19G073100, Dioal.19G074700, and Dioal.19G084600), and two genes (Dioal.19G066100 and Dioal.19G066200) encoding proteins of unknown function that contain C-terminal domains of the ENHANCED DISEASE RESISTANCE 2 (EDR2) family that are negative regulators of plant-pathogen response[60,61]. These QTL are candidates for use in marker-assisted breeding and provide leads for further molecular characterization of anthracnose disease response in yam. However, since variation in levels of infestation, overall plant vigor, and timing and amount of rainfall influence disease severity in field trials, validation of these QTL is required.

In detached leaf assays of the TDa1419 population, performed under varying conditions over three years (Methods), we found a QTL of smaller effect (7.3% of phenotypic variance) on chromosome 6 (Supplementary Fig. 12d–g). While this QTL was marginally significant ($p = 1.28 \times 10^{-2}$), it was found only using three-year averages, and the locus was not significantly associated with YAD in the data from individual years. Furthermore, anthracnose disease levels, as measured by detached leaf assay, were not significantly correlated across genotypes over

**Table 3 Significant QTL identified in this study.**

| Pop. ID | Trait | QTL peak position | n | p-value | Variant | h² | Significance Window[a] |
|---|---|---|---|---|---|---|---|
| TDa1402 | Anthracnose susceptibility (Field 2017) | Chr5: 22,308,637 | 53 | $1.69 \times 10^{-4}$ | A/A,A/G,G/G | 0.4820 | 21,931,073 22,825,712 |
| TDa1402 | Anthracnose susceptibility (Field 2018) | Chr19: 8,369,514 | 49 | $1.25 \times 10^{-2}$ | T/T,T/C | 0.2986 | 3,732,307 17,565,140 |
| TDa1419 | Anthracnose DLA 3-yr mean | Chr6: 61,001 | 243 | $1.28 \times 10^{-2}$ | C/C,C/T | 0.0734 | 38,157 1,418,849 |
| TDa1419 | Dry matter | Chr18: 25,069,928 | 150 | $2.27 \times 10^{-2}$ | C/C,C/T | 0.1020 | 24,779,355 25,415,124 |
| TDa1419 | Oxidation after 30 min[b] | Chr18: 26,496,992 | 151 | $5.86 \times 10^{-3}$ | T/T,T/A,A/A | 0.1367 | 26,199,630 26,749,589 |
| TDa1419 | Oxidation after 180 min[b] | Chr18: 26,496,992 | 151 | $1.38 \times 10^{-2}$ | T/T,T/A,A/A | 0.1188 | 26,199,630 26,749,589 |
| TDa1427 | Oxidation after 30 min | Chr18: 24,495,033 | 97 | $4.52 \times 10^{-6}$ | A/A,A/G | 0.3127 | 24,034,264 24,938,398 |
| TDa1401B | Tuber size | Chr12: 310,852 | 53 | $4.19 \times 10^{-2}$ | T/T,T/C,C/C | 0.2894 | 76,400 489,583 |
| TDa1512 | Tuber shape | Chr7: 3,115,608 | 43 | $3.17 \times 10^{-2}$ | A/A,A/G | 0.3406 | 1,798,899 5,707,988 |

*Pop. ID* mapping population identifier, *n* the number of genotyped and phenotyped progeny used in QTL analysis, *p-value* empirical significance ($\alpha = 0.05$) of the genotype-phenotype association at the peak locus, calculated by Wald statistic-based logistic regression and corrected for family-wise multiple testing by the max(*T*) method, *Variant* alleles segregating at QTL peak position, *h²* narrow-sense heritability.
[a]Calculated as haplotypic linkage disequilibrium ≥0.9 relative to the peak QTL marker.
[b]Same QTL for both oxidation time points in TDa1419.

years. These observations suggest that variation in YAD may be dominated by non-genetic factors.

While previous studies identified two significant anthracnose QTL using EST-SSRs[25] and three QTL using GBS-SNPs[62], none of these colocalize with the QTL in our study. This discrepancy (and the variability seen among different years in our work) may be due to differences in the parental yam genotypes, differences in anthracnose strain and/or inoculation rate in these field studies, and possible genotype-by-environment interactions. Although our parental lines show evidence suggesting past introgression (see below), we did not find any overlaps between these putatively introgressed blocks and our QTL, as might be expected if disease resistance was brought into cultivated yam from a related wild species.

**QTL for tuber quality traits**. Post-harvest oxidation causes browning of yam tuber flesh and flavor changes that reduce crop value[63]. We found an additive-effect QTL for tuber oxidation after peeling at both 30 min ($p = 5.86 \times 10^{-3}$) and 180 min ($p = 1.38 \times 10^{-2}$) on chromosome 18 in the TDa1419 population (Supplementary Fig. 13a–f). The QTL explained 13.67% and 11.88% of the phenotypic variance at 30 and 180 min after peeling, respectively. In the TDa1427 population, a closely linked QTL ($p = 4.52 \times 10^{-6}$), located 2 Mb upstream on the same chromosome, explained 31.3% of the phenotypic variance in oxidation after 30 min (Supplementary Fig. 13g–i). Although enzymatic browning in yam remains poorly understood, polyphenol oxidases and peroxidases are active during browning of *D. alata* and *D. rotundata*[64], and inhibition of this activity has been shown to reduce browning in Chinese yam (*D. polystachya*)[65]. We find a cluster of three peroxidase-encoding genes (Dioal.18G098800, Dioal.18G099400, and Dioal.18G100900) on chromosome 18 at 26.23–26.36 Mb, within ~200 kb of the oxidation QTL at 26.50 Mb in TDa1419 and within 2 Mb of the oxidation QTL in TDa1427, raising the possibility that oxidation is affected by genetic variation in peroxidase activity.

Dry matter (principally starch) content is an important measure of yam yield[66]. We found a single, minor QTL (explaining 10.2% of the phenotypic variance for the dry matter)

on chromosome 18 (Supplementary Fig. 13j–l) in population TDa1419, at position Chr18:25,069,928 ($p = 2.27 \times 10^{-2}$), with genotypes segregating in the population in a pseudo-testcross configuration. Lastly, we identified two QTL for tuber size ($p = 4.19 \times 10^{-2}$) and shape ($p = 3.17 \times 10^{-2}$) in populations TDa1401B and TDa1512, respectively, accounting for 28.9% and 34.1% of their phenotypic variances (Supplementary Fig. 13m–r). While three loci associated with dry matter content and two associated with oxidative browning were previously identified via a genome-wide association study (GWAS)[67], these QTL do not colocalize with those found here, which may be due to differences in the parental yam genotypes or possible genotype-by-environment interactions.

**Genetic variation within *D. alata***. To enable future genetic analyses, we developed a catalog of nearly 3.05 million biallelic single-nucleotide variants (SNVs) in *D. alata*, based on whole-genome shotgun resequencing (Supplementary Note 7, Supplementary Data 1, Supplementary Fig. 14) of breeding lines representing the seven parents of our biparental mapping populations and an additional breeding line (TDa95-310). Of the 3.05 million biallelic SNVs, in our collection, 1.89 million could be confidently genotyped across all individuals. Included within the larger set are 305.5k coding SNVs (251.5k in the reduced set) with predicted effect, 127.1k of which introduce nonsynonymous amino acid changes.

We used these dense SNVs to determine the relationships among the eight breeding lines (Fig. 4a, Supplementary Table 1, Supplementary Data 3) by estimating the fractions of their genomes they shared as identical by descent (IBD). We identified six parent-child relationships (i.e., IBD1, one haplotype shared across the entire genome; relatedness coefficients ~0.50) and five second-degree relationships (i.e., coefficients of ~0.25). All second-degree relations showed unusually high values of IBD1, and both first- and second-degree relations shared substantial IBD2, suggesting a history of recent inbreeding. The relationships inferred are consistent with available pedigree records (Supplementary Table 1), with the addition of several previously unrecorded grandparent-grandchild relationships. Although the

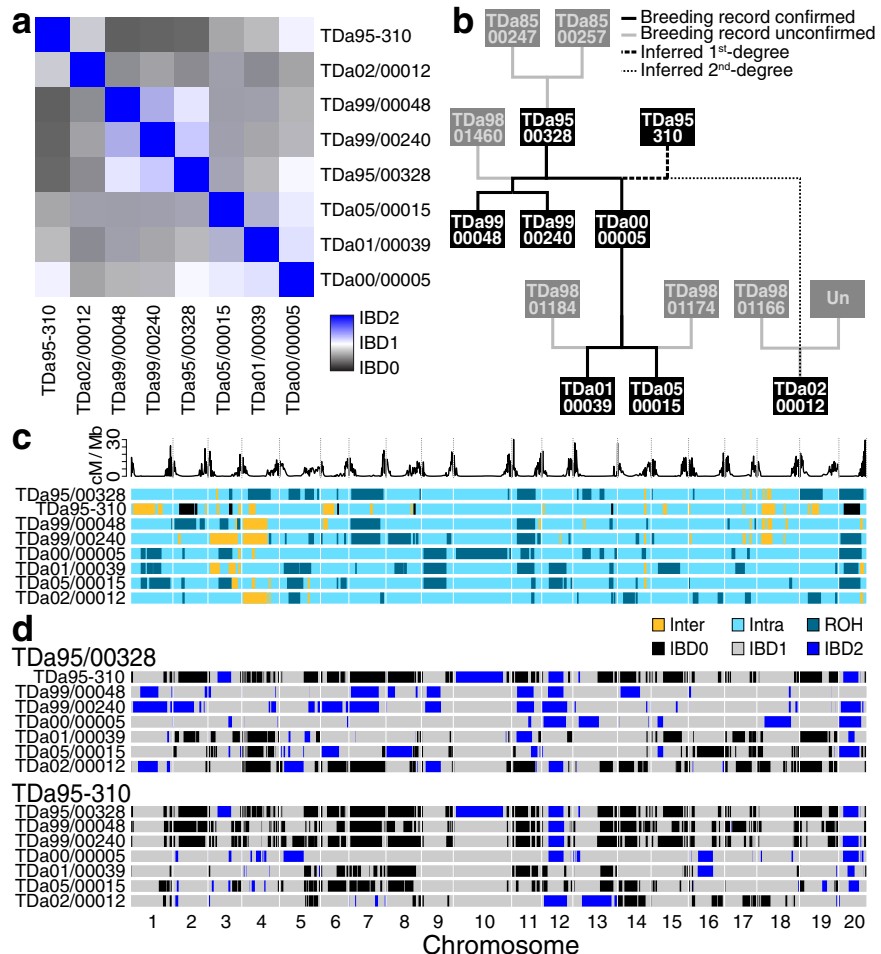

**Fig. 4 Relationships between eight deeply sequenced *D. alata* breeding lines. a** Matrix of identity-by-descent (IBD) relatedness between all pairs of individuals (Supplementary Data 3). Blue represents the degree of diploid genome identity (IBD2); white, one haplotype (IBD1); and black, no shared haplotypes (IBD0). **b** Pedigree of relationships. Sequenced individuals are represented in black boxes with white text, while individuals not sequenced are in gray. Relationships known via IITA records (Supplementary Table 1) are drawn with solid lines. Relationships that could be confirmed using direct sequence comparison are highlighted with solid black lines, and those that could not be are colored grey. Inferred cryptic relationships are indicated with broken lines (first- and second-degree relations are represented as thick dashed and thin dotted lines, respectively). Unexpectedly, TDa95-310 is a parent of TDa00/00005 and a likely second-degree relative of TDa02/00012. **c** Regions of heterozygosity, autozygosity, and possible introgression. Within a background of intraspecific genetic variation (light cyan), large homozygous blocks (runs of homozygosity [ROH], dark cyan) appear common in the resequenced individuals, suggesting autozygosity from historical inbreeding. In addition, large blocks of exceptionally high heterozygosity (yellow) can also be observed, indicating possible introgressions (interspecific variation introduced via hybridization) in one or more of the unsampled pedigree founders. The recombination rate along each chromosome is shown in the track above. **d** Haplotype sharing between TDa95/00328 and all other resequenced individuals, and TDa95-310 and all others. Regions of the genome where an individual shares two haplotypes (i.e., they are IBD2) with TDa95/00328 (or TDa95-310) are highlighted in blue, one shares haplotype (IBD1) in gray, or shares no haplotypes (IBD0) in black. Source data are provided as a Source Data file.

use of highly related parents in breeding programs limits the diversity of alleles available for selection, we note that, as a practical matter, yam crosses are limited to genotypes that flower appropriately, consistently, and profusely.

Unexpectedly, our identity-by-descent analysis shows that TDa95-310 shares a parent-child relationship to TDa00/00005 and a grandparent-grandchild relationship to TDa01/00039 and TDa05/00015. This finding implies that TDa95-310 and the individual TDa98/00150, which appears in the corresponding position in pedigrees, are clones, or that TDa98/00150 is not a parent of TDa00/00005. TDa95-310 is a landrace from Cote d'Ivoire that is likely derived from an accession known as 'Brazo-Fuerte' ('strong arm') introduced from Latin America. It is susceptible to anthracnose and has been used as parent material for crossing[68,69]. We find that TDa95-310 is a second-degree

relative of TDa02/00012. Based on the reported pedigree (Fig. 4b), TDa95-310 must be (1) a parent of either (a) TDa98/01166 or (b) the unknown pollen parent of TDa02/00012, or (2) TDa95-310 also shares one of them as parents. Additional genotyping will resolve this mystery and prevent accidental inbreeding using TDa95-310.

We find extended runs of homozygosity among our eight sequenced lines, as expected based on their high degree of relatedness (Fig. 4c). Long blocks of homozygosity generally stretch across pericentromeric regions, consistent with the low-recombination rates in these regions (Figs. 1 and 4). Although our sampling is not random, the extensive homozygosity (and identity across genotypes) suggests that there may have been selection for the haplotype on chromosome 20 that appears in a homozygous state in six of our eight breeding lines, as well as some other common haplotypes seen in Fig. 4d. The reduced

genetic variation present in these breeding lines suggests a strong need for the introduction of additional diversity in yam breeding programs at IITA and other national institutes.

Conversely, we find that multiple genomes contain several long runs of unusually high heterozygosity (Fig. 4c, Supplementary Fig. 4). While the typical rate of single-nucleotide heterozygosity across 100 kb blocks is ~7–10 SNVs per kb (excluding runs of homozygosity), these highly heterozygous runs have more than 17.5 SNVs/kb (Supplementary Fig. 4c, d, f–g). In cassava and citrus, blocks of high heterozygosity exceeding 10 SNVs/kb variation have been demonstrated to be due to interspecific introgression[70,71]. The co-cultivation of related yam species (Supplementary Fig. 15, Supplementary Note 8, Supplementary Data 4) by growers and breeders suggests that these blocks (some of which are found overlapping low-recombination-rate pericentromeric regions, e.g., on chromosome 4) are the result of past interspecific introgression. Since the Pacific yam D. nummularia is the only other yam species shown to be interfertile with D. alata[20], we speculate that it is the source of introgression into greater yam breeding lines, possibly before introduction to Africa. The retention of these hybrid sequences in this germplasm suggests that they may confer some possible adaptive advantage, as has been hypothesized in cassava (Manihot esculenta Crantz)[70]. Wolfe et al.[72] showed that Manihot glaziovii Muell. Arg. segments introgressed into and maintained as heterozygous in the cassava genome are associated with preferred traits. In the future, a comparison of these highly heterozygous regions with sequences from related Dioscorea spp. should reveal the source of these interspecific contributions to the greater yam germplasm.

Conclusion. The near-complete and contiguous chromosome-scale assembly of D. alata reported here, along with the associated genetic and genomic resources, opens new avenues for improving this important staple crop. We demonstrated the utility of these resources by finding eight QTL for anthracnose disease resistance and tuber quality traits. The genome sequence and associated resources will facilitate future marker-assisted breeding efforts in this crop. A major hurdle for breeders is the difficulty of making successful crosses in D. alata due to lack of flowering, limited seed set, and differences in flowering time. Genome-enabled methods such as marker-assisted selection, GWAS, and genomic selection will allow breeders to make the most out of each cross and use fewer resources to maintain genotypes that are less likely to be useful. By analyzing the diversity of popular breeding lines, we found that they are highly related and, in some cases, have long runs of homozygosity that reduce the genetic diversity available for selection but may represent genomic regions fixed for desirable traits. Analysis of a broader sampling of African greater yam germplasm will prove valuable to avoiding inbreeding depression associated with inbreeding elite lines[73]. Conversely, we found regions of presumptive interspecific hybridization, pointing to the potential value of broader crosses that may enable the transfer of valuable traits from other yam species while minimizing linkage drag with genome-assisted selection. Similarly, the genome sequence also enables the application of gene editing to directly alter genotypes in a targeted manner, preserving genetic backgrounds that confer cohorts of desirable traits. The small genome of D. alata and the advent of rapid long-read technologies open the door to rapidly assemble additional accessions to discover and leverage structural variants for breeding. Such variants have been shown to control important traits, such as plant development[74], and contribute to reproductive isolation[75].

Greater yam has a high potential for increased yield and broader cultivation, with advantages compared with other root-tuber-banana crops due to its superior nutritious content and low glycemic index[76,77]. Greater yam's ability to grow in tropical and sub-temperate regions around the world suggests that it is highly adaptable to its environment and that there may be adaptive traits (and associated alleles) that could be exploited in different global contexts. It establishes itself vigorously, is higher yielding than other domesticated yam species, and is highly tolerant to marginal, poor soil and drought conditions, and thus likely nutrient use efficient[8]. These traits will be valuable assets in a changing climate. Greater yam is also highly tolerant of the most significant yam virus, yam mosaic virus[19]. By leveraging QTL and genome-wide association for disease resistance and tuber quality, as well as marker-aided breeding strategies and genome editing, yam breeders are poised to rapidly generate disease-resistant, high-performing, farmer-/consumer-preferred, climate-resilient varieties of greater yam.

## Methods

Reference accession. The breeding line TDa95/00328, from the International Institute of Tropical Agriculture (IITA) yam breeding collection, was chosen as the D. alata reference genome accession because it is moderately resistant to anthracnose (a fungal disease caused by Colletotrichum gloeosporioides) and was confirmed to be diploid by marker segregation analysis[23,27]. Chromosome number ($2n = 40$) was further confirmed through chromosome counting (Supplementary Note 1, Supplementary Fig. 2).

Genome sequencing. High molecular weight DNA for Pacific Biosciences (PacBio, Menlo Park, USA) Single-Molecule Real-Time (SMRT) continuous long-read (CLR) sequencing was isolated as described in Supplementary Note 1. PacBio library preparation and sequencing were performed at the University of California Davis Genome and Biomedical Sciences Facility. Three libraries were constructed as per manufacturer protocol, with fragments smaller than 7, 15, and 20 kb, respectively, excluded using Blue Pippin. In total, one RSII and 20 Sequel SMRT cells of CLR data were generated for a combined 235× sequence depth. Half of the 112.4 Gb of generated bases were sequenced in reads 14.5 kb or longer.

For HiC chromatin conformation capture, suspensions of intact nuclei from D. alata (TDa95/00328) were prepared from young leaves and apical parts of the stem according to ref. [78]. at the Institute of Experimental Botany, Olomouc, Czech Republic, with modifications as described in Supplementary Note 1. These nuclei were sent to Dovetail Genomics for HiC library preparation[79], which were sequenced on an Illumina HiSeq 4000 to produce 358.5 million 151 bp paired-end reads.

For genome sequence polishing, a 625 bp insert-size Illumina TruSeq library was made and sequenced on a HiSeq 2500 at UC Berkeley's Vincent J. Coates Genomics Sequencing Lab (VCGSL), yielding 131 million 251 bp paired reads (137× depth). For contig linking, three Nextera mate-pair libraries (insert sizes ~2.5 kb, 6 kb, and 9 kb) were prepared and sequenced as 151 bp paired-end reads on a HiSeq 4000 at the UC Davis Genome and Biomedical Sciences Facility. More details are described in Supplementary Note 1. A listing of all TDa95/00328 sequencing data, and corresponding NCBI Sequence Read Archive (SRA) accession numbers, may be found in Supplementary Data 1.

Genome assembly. We assembled the D. alata genome sequence with Canu[80] v1.7-221-gb5bffcf from the longest 110× of PacBio CLR reads (50.228 Gb in reads 19.8 kb or longer). Contigs were filtered down to a single mosaic haplotype in JuiceBox[81,82] v1.9.0, considering median contig depth (Supplementary Fig. 3), sequence similarity, and HiC contacts. Non-redundant contigs were scaffolded into chromosomes using SSPACE[83] v3 and 3D-DNA[84] commit 2796c3b. Misassemblies were corrected manually with the aid of genetic maps and JuiceBox HiC visualization. The assembly was polished twice with Arrow[85] v2.2.2 (SMRT Link v6.0.0.47841) followed by two rounds of Illumina-based polishing with FreeBayes[86] v1.1.0-54-g49413aa and custom scripts (Supplementary Note 1).

DArTseq genotyping. DNA was isolated at IITA and NRCRI from their respective mapping populations and parents using modified CTAB methods (Supplementary Note 2). DNA samples were genotyped by Integrated Genotyping Service and Support (IGSS, BecA-ILRI hub, Nairobi, Kenya) or DArT (Canberra, Australia) using the 'high-density' DArTseq reduced-representation method. DArTseq genotype datasets were deposited in Dryad [https://doi.org/10.6078/D1DQ54][87]. Lists of sequence data used for DArTseq genotyping, and corresponding NCBI Sequencing Read Archive (SRA) accession numbers, are provided in Supplementary Data 1.

Genetic linkage mapping. DArTseq genotyping datasets were mapped to the v2 genome sequence, then filtered for a minimum 90% genotyping completeness and $F_1$ Mendelian segregation via $\chi^2$ goodness-of-fit tests ($\alpha = 1 \times 10^{-2}$) on allele and

genotype frequencies using MapTK[88] v1.4.1-11-g19a5f3a (https://bitbucket.org/rokhsar-lab/gbs-analysis) and VCFtools[89]. Half-sibs, off-types, and sample errors were detected via clustering as in ref. [88]. and removed. Parental genotypes from one dataset were substituted when a sample by the same name was found to be inconsistent in another. Genotypes were phased and imputed using AlphaFamImpute[90] v0.1 and parent-averaged linkage maps constructed in JoinMap[91,92] v4.1 with the maximum-likelihood mapping function for cross-pollinated populations, which were then integrated into a composite map using LPmerge[93] v1.7. Further detail regarding genetic linkage mapping can be found in Supplementary Table 2 and Supplementary Note 2. All linkage maps were deposited in Dryad [https://doi.org/10.6078/D1DQ54][87].

**RNA sequencing**. RNA was extracted at ICRAF from 12 tissues from a single TDa95/00328 plant grown onsite in Nairobi, Kenya. Tissues included leaf petiole, roots, various stages of leaves (initial sprouting leaf, leaf bud, young leaf, semi-matured leaf, matured leaf, fifth leaf), bark, stem, first internode, and middle vine as described in Supplementary Note 3. RNA samples were pooled for sequencing by two technologies.

Illumina RNA-seq libraries were prepared using the TruSeq stranded mRNA preparation kit (Illumina cat# 20020594) and sequenced at the Agricultural Research Council Biotechnology Platform (ARC-BTP) in Pretoria, South Africa on an Illumina HiSeq 2500 as 125 bp paired ends (SRA: SRR13683865 [https://www.ncbi.nlm.nih.gov/sra/SRR13683865]).

Oxford Nanopore Technologies (ONT) Direct-RNA Sequencing (Nanopore DRS) and data processing were performed at the University of Dundee, Dundee, UK. The Nanopore DRS library was prepared using the SQK-RNA001 kit (ONT)[94], using 5 μg of total RNA as input for library preparation, and sequenced on R9.4 SpotON Flow Cells (ONT) using a 48 h runtime. Nanopore DRS reads (SRA: SRR13683864) were base-called using Guppy v2.3.1 (ONT), then corrected using proovread[95] v2.14.1 without sampling. Transcript assemblies were generated with Pinfish (ONT) v0.1.0 from corrected reads aligned to the v2 genome sequence with Minimap2 v2.8 (ref. [96]). More details on Nanopore transcriptome sequencing are in Supplementary Note 3.

**Protein-coding gene annotation**. Transcript assemblies (TAs) were constructed with PERTRAN[97] v2.4 from 107 M pairs of Illumina RNA-seq reads, combining our data with those from Wu et al.[98] (SRA: SRR1518381 and SRR1518382) and Sarah et al.[99] (SRA: SRR3938623) along with 44k 454 ESTs from Narina et al.[68] (SRA: SAMN00169815, SAMN00169801, SAMN00169798). A merged set of 86,399 TAs were constructed by PASA[100] v2.0.2 from the above RNA-seq TAs along with 53k assemblies from corrected Nanopore DRS reads, and 18 full-length cDNAs collected from NCBI.

Protein-coding genes were predicted with the DOE-JGI Integrated Gene Call[101] (IGC) v5.0 annotation pipeline, which integrates TA evidence and ab initio gene predictions. Briefly, gene loci were determined by TA alignments and/or EXONERATE[102] v2.4.0 peptide alignments from *Arabidopsis thaliana*[39] TAIR10, *Glycine max*[103] Wm82.a4.v1, *Sorghum bicolor*[104] v3.1.1, *Oryza sativa*[105] v7.0, *Setaria viridis*[106] v2.1, *Amborella trichopoda*[107] v1.0, *Zostera marina*[108] v2.2, *Musa acuminata*[109] v1, *Ananas comosus*[51] v3, and *Vitis vinifera*[110] v2.1 proteomes obtained from Phytozome[111] v13 (https://phytozome-next.jgi.doe.gov) and Swiss-Prot[112] proteins (2018, release 11). Gene models were predicted using FGENESH + [113] v3.1.1, FGENESH_EST v2.6, PASA (v2.0.2) assembly-derived ORFs, and AUGUSTUS v3.3.3 via BRAKER1 v1.9 (ref. [114].). After selecting the best-scoring predictions at each locus (Supplementary Note 3), UTRs and alternative transcripts were added with PASA. Functional annotations were predicted with InterProScan[115] v5.17-56.0. The annotation completeness of this and other Dioscoreaceae species (Supplementary Table 5) were measured using BUSCO[31] v3.0.2-11-g1554283 with the Embryophyta OrthoDB[32] v10 database.

**Genomic repeat annotation**. Repeat annotation was performed twice (see Supplementary Note 3) with RepeatMasker[116] v4.1.1. The initial round annotated de novo repeats inferred from the preliminary v1 assembly by RepeatModeler[117] v1.0.11, combined with *Dioscorea* repeats deposited in RepBase[118]. The second round used a repeat library inferred by RepeatModeler v2.0.1 (-LTRstruct) from the more complete v2 genome sequence.

**Comparisons with other monocot genomes**. Orthologous genes were clustered with OrthoFinder[119] v2.4.1 across the available assembled Dioscoreaceae species: *D. alata*, *D. rotundata*[21] (GCA_009730915.1), *D. dumetorum*[34] (GCA_902712375.1), *D. zingiberensis*[22] (GCA_014060945.1), and *Trichopus zeylanicus*[35] (GCA_005019695.1). This procedure produced 5,454 clusters of genes in strict 1:1:1:1 correspondence among the *Dioscorea* species of which 99.9% (n = 5451), 90.5% (n = 4937), and 99.1% (n = 5404) were localized to chromosome-scale scaffolds in *D. alata*, *D. rotundata*, and *D. zingiberensis*, respectively. We also used OrthoFinder to compare a broader set of monocots (*D. alata*, *D. rotundata*, *D. dumetorum*, *D. zingiberensis*, *T. zeylanicus*, *Xerophyta viscosa*[120] (GCA_002076135.1), *Apostasia shenzhenica*[121] (GCA_002786265.1), *Dendrobium catenatum*[122] (GCF_001605985.2), *Asparagus officinalis*[53] (GCF_001876935.1), *Elaeis guineensis*[52] (GCF_000442705.1), *Phoenix*

*dactylifera*[123] (GCF_000413155.1), *Musa acuminata*[109] (GCF_000313855.2), *Oriza sativa*[124] (GCF_001433935.1), *Zea mays*[125] (GCF_000005005.2), *Ananas comosus*[51] (GCF_001540865.1), *Spirodela polyrhiza*[54,126] (GCA_000504445.1, GCA_001981405.1), *Zostera marina*[108] (GCA_001185155.1)) with *Arabidopsis thaliana*[39,127] (GCF_000001735.4) and *Amborella trichopoda*[107] (GCF_000471905.2) as outgroups. These results are presented graphically in Supplementary Fig. 8 using the ClusterVenn[128] online tool (https://orthovenn2.bioinfotoolkits.net/cluster-venn). See Supplementary Note 3 and Supplementary Data 4 for more detail.

**Chromosome landscape, Rabl chromatin structure, and centromere estimates**. The A/B compartment structure (Supplementary Fig. 7) for each chromosome was inferred at 100 kb resolution with Knight-Ruiz (KR)-balanced MapQ30 intra-chromosomal HiC count matrices using a custom script (call-compartments v0.1.2-67-g18fff4a; https://bitbucket.org/bredeson/artisanal). Centromeric positions were estimated in JuiceBox (v1.9.0) following the principles described by Varoquaux et al.[129]. Rabl chromatin structure (Supplementary Note 4) was extracted in R[130] v3.5.3 using the prcomp function (chr-structure.R v1.0; https://github.com/bredeson/Dioscorea-alata-genomics) on KR-balanced MapQ30 inter-chromosomal HiC count matrices, with chromosome 2 as the reference comparator. Pearson's correlations (r) between gene count, low-complexity and transposable element repeat densities, recombination rate, and A/B compartment domain status were computed using 500 kb non-overlapping windows with BEDtools[131] v2.28.0 and R[130] v3.5.3 (Supplementary Note 4). Putative centromere sequences and loci (Supplementary Data 2) were determined using a combination of HiC and tandem-repeat finding approaches (Supplementary Note 4).

**Synteny and comparative genomics**. We used BLASTP[132,133] (BLAST + v2.10.0) to search for homologous proteins between *Dioscorea alata* and each comparator species: *Ananas comosus*[51] (GCF_001540865.1), *D. rotundata*[21] (GCA_009730915.1), *D. dumetorum*[34], *D. zingiberensis*[22] (GCA_014060945.1), *Elaeis guineensis*[52] (GCF_000442705.1), *Spirodela polyrhiza*[54,126] (GCA_000504445.1, GCA_001981405.1), and *Trichopus zeylanicus*[35] (GCA_005019695.1). DIALIGN-TX[134] v1.0.2 and the kaks function from the SeqinR[135] v3.6-1 R[130] (v3.5.3) package were used to calculate synonymous substitution ($K_S$) rates. Runs of collinear loci (Supplementary Data 2) were inferred using custom filtering and clustering scripts (run-collinearity.sh v1.0, https://github.com/bredeson/Dioscorea-alata-genomics; cluster-collinear-bedpe v0.1.2-67-g18fff4a, https://bitbucket.org/bredeson/artisanal). See Supplementary Note 5 for more details. All ribbon diagrams were generated with the jcvi.graphics.karyotype module in MCscan[136] v1.0.14-0-g58b7710b.

**Mapping populations at IITA**. Phenotyping of five mapping populations was performed at IITA from 2016–2019. In 2016, mapping populations were planted in single pots and grown in the screenhouse for seed tuber multiplication and screening of anthracnose disease in a controlled environment. In 2017, individual mini-tubers of each mapping population were pre-planted in pots to ensure germination, and one-month-old seedlings were transplanted in the field using a ridge-and-furrow system. Land preparation, weeding, staking and harvesting were carried out following standard field operating protocol for yam[137]. In 2018 and 2019, harvested tubers were cut into mini-sets of 100 g each, treated with pesticide to prevent rotting, and planted in the field as above. More detail on the planting scheme used at IITA may be found in Supplementary Note 6.

**Phenotyping for anthracnose disease**. Populations were assessed for yam anthracnose disease (YAD) at the International Institute for Tropical Agriculture (IITA, Ibadan, Nigeria) and the National Root Crops Research Institute (NRCRI, Umudike, Nigeria). More detailed descriptions of phenotyping for YAD may be found in Supplementary Note 6; all YAD phenotyping datasets were deposited in Dryad [https://doi.org/10.6078/D1DQ54][87].

For the five IITA populations (TDa1401, TDa1402, TDa1403, TDa1419 and TDa1427), each plant was visually scored in the field in 2017 and 2018 for YAD severity at 3 months after planting (MAP) and 6 MAP using a 1–5 scale as follows: Score 1 = No symptoms, Score 2 = 1–25%, Score 3 = 25–50%, Score 4 = 50–75%, Score 5 ≥ 75%. Detached leaf assays (DLA) were performed at IITA on plants grown in the screenhouse in 2016, and on plants grown in the field in 2017 and 2018, following a modified protocol of Green et al.[138] and Nwadili et al.[139].

At NRCRI, site-specific *C. gloeosporioides* isolates were collected and evaluated, as described in Supplementary Note 6. The most virulent isolate was used for anthracnose severity evaluation of NRCRI *D. alata* mapping populations using DLA[139].

**Phenotyping for post-harvest tuber traits**. Tuber dry matter content was phenotyped at IITA. After harvest, healthy yam tubers were sampled in each replication for dry matter determination. The tubers of each genotype were cleaned with water to remove soil particles. Thereafter, the tubers were peeled and grated for easy oven drying; 100 g of freshly grated tuber flesh sample was weighed, put into a Kraft paper bag, and dried at 105 °C for 16 h. After drying, the weight of each

sample was recorded and the dry matter content was determined using Eq. 1:

$$\% \text{ Dry matter content} = 100 \cdot \frac{\text{weight of dry sample }(g)}{\text{weight of fresh sample }(g)} \quad (1)$$

Tuber flesh color and oxidation/oxidative browning were phenotyped at IITA. After harvest, one well-developed and mature representative tuber was sampled in each replication. The sampled tuber was peeled, cut, and chipped with a hand chipper to get small thickness size pieces. A chromameter (CR-410, Konica Minolta, Japan) was used to read the total color of sampled pieces placed on a petri dish immediately and exposure to air at 0, 30, and 180 min. The lightness ($L^*$), red/green coordinate ($a^*$), and yellow/blue coordinate ($b^*$) parameters were recorded for each chromameter reading for the determination of the total color difference. A reference white porcelain tile was used to calibrate the chromameter before each determination[140]. Tuber whiteness was calculated with Eq. 2:

$$f_L = \frac{L^{*2}}{L^{*2} + a^{*2} + b^{*2}} \quad (2)$$

where $\Delta L^*$ = difference in lightness and darkness ([+] = lighter, [−] = darker), $\Delta a^*$ = difference in red and green ([+] = redder, [−] = greener), and $\Delta b^*$ = difference in yellow and blue ([+] = yellower, [−] = bluer) (http://docs-hoffmann.de/cielab03022003.pdf).

Tuber flesh oxidation was estimated from the total variation from the difference in the final and initial color reading, as in Eq. 3:

$$\text{Tuber flesh oxidation} = E_{\text{final}} - E_{\text{initial}} \quad (3)$$

where $\Delta E_{\text{final}}$ = color reader value at the final time (30 min) and $\Delta E_{\text{initial}}$ = Initial color reader value (at 0 min).

Tubers were evaluated post-harvest at NRCRI. Of the three populations evaluated at NRCRI, 172 progeny survived. As soon as the yam tubers were harvested, eight traits were assessed using the descriptors from Asfaw[137]: presence or absence of corm (CORM: 0 = absent; 1 = present), the ability of corm to separate (CORSEP: 0 = no; 1 = yes), type of corm (CORTYP: 1 = regular; 2 = transversally elongated; 3 = branched), tuber shape (TBRS: 1 = spherical/round; 2 = oval; 3 = cylindrical; 5 = irregular), tuber size (TBRSZ: 1 = small, length less than 15 cm; 2 = medium, length between 15 and 25 cm; 3 = big, length longer than 25 cm), tuber surface texture (TBRST: 1 = smooth; 2 = rough), roots on tuber (RTBS: 0 = no roots; 2 = few; 3 = many) and position of roots on tuber (PRTBS: 1 = lower; 2 = middle; 3 = upper; 4 = entire tuber). Tuber trait phenotyping datasets for all mapping populations were deposited in Dryad [https://doi.org/10.6078/D1DQ54][87].

**QTL analysis.** QTL association analyses integrated linkage maps, imputed genotype data, and phenotype data into Binary PED files using PLINK[141,142] v1.90b6.16. Only progeny samples with both genotype and phenotype data were retained per trait. Some traits were initially scored using a discrete 0–2 system, which PLINK assumes are missing/case/control phenotypes; these trait values were shifted out of the 0–2 range before analysis by adding an offset of 1 or 2 to all values (depending on initial data range). An independent QTL association analysis was performed for each trait using logistic regression. Per-locus Wald statistic $p$-values were adjusted for multiple testing by $\max(T)$ correction[141,143] with $1 \times 10^6$ phenotype label-swap permutations. A locus was considered significant if the empirical $\max(T)$-corrected $p$-value was less than $\alpha = 0.05$. Two dry matter phenotype measurements were excluded from the TDa1419 population: TDa1419_485 (a likely typographical error in data collection) and TDa1419_142 (an extreme outlier value).

For each identified QTL, an effect plot was generated to determine the dominance pattern and estimate narrow-sense heritability ($h^2$) at the peak marker. Effect plots and $h^2$ were calculated as described by Broman and Sen[144] (pg. 122) using a custom R[130] script (plot-qtl-gxp.R v1.0, https://github.com/bredeson/Dioscorea-alata-genomics). The effect status (i.e., dominance) for chromosomes 6 and 19 anthracnose QTL could not be determined because the alleles at these loci are segregated in pseudo-testcross configurations. The interval around each QTL peak (Table 3) was determined by expanding the interval boundaries upstream and downstream of the peak marker until another marker with linkage disequilibrium (LD) below 0.9 was encountered (plot-qtl-ld.R v1.0, https://github.com/bredeson/Dioscorea-alata-genomics). The gene loci contained within these intervals, and their functional annotations, are provided in Dryad [https://doi.org/10.6078/D1DQ54][87]. In addition to the predicted functional annotations (Supplementary Note 3) for each *D. alata* gene, protein descriptions were included from the best BLASTP[133] (-seg yes -lcase_masking -soft_masking true -evalue 1e-6) hits to the NCBI RefSeq proteomes (release 207, 2021-07-15) of *Arabidopsis thaliana*, *Gossypium hirsutum*, *Ipomoea batatas*, *Malus domestica*, *Medicago truncatula*, *Musa acuminata*, *Nicotiana tabacum*, *Oryza sativa Japonica*, *Solanum lycopersicum*, *Solanum tuberosum*, *Vitis vinifera*, and *Zea mays* when searching for causal gene candidates within QTL intervals.

**WGS Illumina sequencing.** DNA samples from the breeding lines listed in Supplementary Table 1 were isolated at IITA (Supplementary Note 7). TruSeq Illumina libraries were constructed and sequenced at the VCGSL. Inferred insert sizes ranged from 247–876 bp. These libraries were sequenced on HiSeq 2500 or HiSeq 4000 with reading lengths ranging from 150–251 bp, yielding combined sample

depths of 19–230×. Supplementary Data 1 lists all Illumina sequence data from our breeding lines, including external data, and accompanying summary statistics.

**WGS variant calling.** Single-nucleotide variants (SNVs) were called from the whole-genome resequencing datasets listed in Supplementary Data 1. Briefly, Illumina reads were screened for TruSeq adapters with fastq-mcf (ea-utils[145] tool suite) v1.04.807-18-gbd148d4, then aligned with BWA-MEM[146] v0.7.17-11-g20d0a13 to a TDa95/00328 v2 genome index containing *D. alata* plastid (GenBank: MZ848367.1 [https://www.ncbi.nlm.nih.gov/nuccore/MZ848367.1]) and mitochondrial (GenBank: OK106275.1 [https://www.ncbi.nlm.nih.gov/nuccore/OK106275.1]) sequences and a *Pseudomonas fluorescens* chromosome (GenBank: CP081968.1 [https://www.ncbi.nlm.nih.gov/nuccore/CP081968.1]) as bait for contaminant reads. BAM files were processed with SAMtools[147] v1.9-93-g0ca96a4 to fix mate information, mark duplicates, sort, merge, and filter for properly-paired reads. Initial SNVs and indels were called with the Genome Analysis ToolKit[148] (GATK; v3.8-1-0-gf15c1c3ef) HaplotypeCaller and GenotypeGVCFs tools. False-positive variant and genotype calls were filtered using individual-specific minimum- and maximum-depth cutoffs, allele-balance binomial test thresholds ($\alpha = 0.001$; Supplementary Fig. 14), a read depth mask, and annotated repeat masks. See Supplementary Note 7 for a more complete description of the filtering protocol used. Only biallelic SNVs were used in downstream analyses and effect predictions were annotated with SnpEff[149] v5.0.c2020-11-25.

**WGS population analyses.** Using 1.89 million SNVs with 75% or more of individuals genotyped, pairwise genome-wide relatedness estimates were obtained with VCFtools[89] v0.1.16-16-g954e607. The resulting relatedness network and origination year encoded in each sample's identifier were used to verify IITA pedigrees. The intrinsic heterozygosity and autozygosity of each individual, as well as the pairwise segmental (5000 SNV windows, 1000 SNV step) identity-by-descent (IBD) of each, were estimated with custom tools (snvrate and IBD tools v1.0-26-g4cf73ab, https://bitbucket.org/rokhsar-lab/wgs-analysis). A 100 kb sliding window (10 kb step) was called autozygous if the rate of intrinsic heterozygosity was less than $2 \times 10^{-4}$. This threshold was determined empirically (Supplementary Fig. 4, Supplementary Note 7).

**Mitochondrial and plastid sequence assemblies and phylogenetics.** Mitochondrial and plastid DNA sequences were assembled using de novo and comparative methods (Supplementary Note 8). The IboSweet3 *D. dumetorum* plastid was extracted from the Siadjeu et al.[34] assembly. Our Dioscoreaceae DNA phylogeny was built from plastid long single-copy regions using MAFFT[150,151] FFT-NS-i v7.427 (--6merpair --maxiterate 1000), Gblocks v0.91b, and PhyML[152] v3.3.20190909 (--leave_duplicates --freerates -a e -d nt -b 1000 -f m -o tlr -t e -v e). The monocot plastid phylogeny was constructed using OrthoFinder[119,153,154] v2.4.1 (MAFFT v7.427 alignment and IQ-TREE[155] v2.0.3 phylogenetic reconstruction) single-copy orthologs. All trees were visualized with FigTree v1.4.4 (https://github.com/rambaut/figtree).

**Reporting summary.** Further information on research design is available in the Nature Research Reporting Summary linked to this article.

## Data availability

A reporting summary for this article is available as a Supplementary Information file. The genome sequence, annotation, and SNP data are browsable at Phytozome [https://phytozome-next.jgi.doe.gov/info/Dalata_v2_1] or YamBase [https://yambase.org/organism/Dioscorea_alata/genome]. The *D. alata* TDa95/00328 nuclear genome (GCA_020875875.1), transcriptome (GJIX00000000.1), plastid (MZ848367.1), and mitochondrion (OK106275.1) assemblies, and *Pseudomonas fluorescens* chromosome (CP081968.1) were deposited in the NCBI GenBank database. *D. rotundata* TDr96_F1 and *D. dumetorum* IboSweet3 plastid sequences were also deposited in the NCBI GenBank database under accessions MZ848368.1 and MZ848369.1, respectively. All sequencing read data generated for this work were deposited in the NCBI Sequence Read Archive (SRA) under BioProject PRJNA666450; see Supplementary Data 1 for individual sample SRA metadata. The genetic linkage maps, phenotype datasets, and DArTseq genotype datasets for all populations, as well as functional annotations for all genes within QTL intervals, were deposited in Dryad [https://doi.org/10.6078/D1DQ54][87]. Source Data files are provided with this work. Source data are provided with this paper.

## Code availability

Analysis scripts used throughout this work are available at Github [https://github.com/bredeson/Dioscorea-alata-genomics] (tag 'v1.0') and Bitbucket: [https://bitbucket.org/rokhsar-lab/wgs-analysis] (v1.0-26-g4cf73ab), [https://bitbucket.org/rokhsar-lab/gbs-analysis] (v1.4.1-11-g19a5f3a), and [https://bitbucket.org/bredeson/artisanal] (v0.1.2-67-g18fff4a).

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

## Acknowledgements

At the University of California, Davis, Genome and Biomedical Sciences facility, we thank Oanh Nguyen for troubleshooting and advice for DNA isolation and PacBio sequencing, Emily Kumimoto for mate-pair libraries, and Lutz Froenicke for management. For facilitating DArTseq genotyping, we thank: Andrzej Kilian (Diversity Arrays Technology); and Clay Sneller, Jackline Chepkoech, Mercy Chepngetich, and IGSS/SEQART staff at BecA-ILRI Hub. We thank the staff of Bioscience Center, Yam Breeding Unit, Pathology/Virology Unit, and Farm Office at IITA, Ibadan, Nigeria for support in laboratory and field activities. We thank Kwabena Darkwa and Agre Paterne, IITA, Ibadan Nigeria for their support in phenotyping population TDa1401. Boas Pucker provided the single-haploid assembly of *D. dumetorum*. Christopher Saski and Mary Duke provided WGS data of TDa95/00328 and TDa95-310. We thank Ismail Rabbi for early discussions in proposal development, and he and Gezahegn Girma for providing *D. alata* DNA of specific breeding lines. This work is based on a project supported by the National Science Foundation BREAD program, Award No. 1543967 to D.S.R., R.B., and J.E.O. We wish to acknowledge subsidy from the Integrated Genotyping Service and Support platform, a collaborative project between the International Livestock Research Institute (ILRI) and the Bill and Melinda Gates Foundation. DNA extractions for PacBio sequencing, and RNA extractions, were carried out at ICRAF with partial support from the African Orphan Crops Consortium. RNA-seq was funded by the Illumina Greater Good Initiative. Nanopore DRS work was supported by The University of Dundee Global Challenges Research Fund to G.G.S. and G.J.B., Biotechnology and Biological Sciences Research Council (BB/M004155/1) to G.G.S. and G.J.B. and H2020 Marie Skłodowska-Curie Actions (799300) to K.K. Sequencing performed at the Vincent J. Coates Genomics Sequencing Laboratory, UC Berkeley, was partially supported by NIH S10 OD018174 Instrumentation Grant. D.S.R. was supported by Chan Zuckerberg BioHub, internal funds at the Okinawa Institute of Science and Technology, and the Marthella Foskett-Brown Chair in Biological Science at UC Berkeley. This research used resources of the National Energy Research Scientific Computing Center, which is supported by the Office of Science of the U.S. Department of Energy under Contract No. DE-AC02-05CH11231.

## Author contributions

Conceived, designed, and led study: D.S.R., R.B., J.E.O., J.V.B., J.B.L. Genome assembly and chromatin structure, chromosome landscape, comparative genomics, chromosome evolution, population genetic, and phylogenetic analyses: J.V.B. (lead), D.S.R. Genome sequencing planning and coordination: D.S.R., J.V.B., A.V.D., J.B.L. Genetic mapping: J.V.B. (lead), J.B.L. QTL analysis: J.V.B. Overall project management: J.B.L. Mapping population development: A.L.M., A.A. Mapping population management/propagation: R.B., I.O.O., A.A., J.N., I.N. Development of and info on breeding lines: R.A., A.L.M. Phenotyping of mapping populations: I.O.O., O.K., A.A., P.L.K., N.R.O., C.O.N., I.N., J.N. Preparation of cell nuclei for HiC analysis; karyotype and chromosome counting: J.D. (lead), E.H. Nanopore DRS sequencing and analysis: M.P., K.K., A.V.S., G.J.B., G.G.S. (lead). DNA isolation for reference genome, sequencing of breeding lines, and genotyping: I.O.O., N.R.O., J.N., R.K., S.M., P.S.H. RNA isolation: R.K., S.M., P.S.H. (lead). Provision of RNA-seq data: J.F. Wrote manuscript: D.S.R., J.V.B., J.B.L., J.E.O., O.K., N.R.O., C.N., R.B., E.H. with input from A.V.D., G.G.S., J.D. Annotation and database management: D.G. (lead), S.S., J.C. Other project planning/site-specific supervision: I.O.O., C.N.E., R.J., AM.

## Competing interests

D.S.R. is a member of the Scientific Advisory Board of, and a minor shareholder in, Dovetail Genomics LLC, which provides as a service the high-throughput chromatin conformation capture (HiC) technology used in this study. The remaining authors declare no competing interests.
