## [Peer Review File · Nature Communications]

Chromosome evolution and the genetic basis of agronomically important traits in greater yamReviewers' Comments:

Reviewer #1:

Remarks to the Author:

The manuscript by Bredeson et al. entitled "Chromosome evolution and the genetic basis of agronomically important traits in greater yam" presents a comprehensive and high-quality study centred on the genome sequence of *Dioscorea alata* (greater yam). The genome sequence has been generated based on SMRT long reads (PacBio), has reached chromosome arm quality including quite some telomers (centromeres missing), and has been ordered based on a newly generated high resolution genetic map. In general, a very convincing study!

The analyses regarding chromosome evolution in the *Dioscorea* species yielded interesting results regarding a "delta" and a "tau" WGD, these are well documented in the results (and the comprehensive supplements) and are also nicely put into literature context.

The QTL analyses are substantial and have been performed with quite some resources, however, the results are not that impressive. Although there is one candidate QTL locus on chromosome 5 (see Fig. 3) explaining "48.2% of phenotypic variance in the 2017 data", this QTL was not validated in other years or other population. Given the problems which are correctly mentioned and very appropriately discussed in the paper, the statement in the Concluding Remarks regarding "facilitat[ing] marker-assisted breeding in this crop [*D. alata*]" seems a bit far fetched. Nevertheless, it is very obvious that the genomic and genetic resources provided by this paper will greatly support *D. alata* breeding. The results and data presented are novel and are surely relevant for a larger audience.

Specific points:

1) Throughout: The authors should clearly distinguish between the terms "genome" and "genome sequence". Only a genome sequence can be assembled and annotated. Only the genome (as DNA in the cell) has the ability to become realised in a phenotype or trait. Reading the text is difficult when this distinction is blurred. The problem becomes very obvious in the paragraph "Comparative analysis and paleopolyploidy", but it starts already in the abstract.

Another point also addressing terminology and the use of designations. Throughout the text, it should be either "*D. alata*" OR "greater yam", but not a mix. Jumping between the two terms is confusing.

2) It must be made more obvious to the reader which published sequences were used for comparisons. This information is presented in "Supplementary Table 7" and is mentioned here and there in the supplements, but the references and identifiers must be presented clear and unambiguous also in the main text (e.g. page 7 line 178). It is not sufficient to add more pointers to "Supplementary Table 7".

3) page 4 line 90: Citation of "Gatarira et al., in preparation)". It is not clear to this reviewer why this citation is included here. Supplementary Fig. 2 shows the chromosome count ($2n = 40$) for TDa95/00328, the sequenced reference genotype. The Supplementary Methods provide the respective protocol as required. If this citation is relevant, at least a preprint (at bioRxiv) needs to be provided and cited.

4) page 5 line 99: The text regarding the populations used to generate the genetic maps is difficult to follow. It should be made clear why there are different numbers mentioned at different places in the text:

- line 112: "ten genetic maps were highly concordant"

- Legend to Figure 1b: composite genetic linkage map (black points), integrating five mapping populations

- Table 2: 9 parent pairs listed, but 11 population IDs, and TDa1402 & TDa1506 & TDa1621 as well as TDa1419 & TDa1610 share identical parents

This reviewer assumes that rephrasing the text should clarify the questions marks.

5) page 8 line 214: What is the evidence for $n=20$ for *D. dumetorum*? Neither data nor a reference is

provided. It might be that the old estimation of $n=18/2n=36$ (Miege, *Revue de Cytologie et de Biologie Végétales*, 1954) is not correct. But if so, the authors need to present evidence for the correction.

6) The sequence data for TDa95-310 and TDa95/00328 must be submitted as well (see Supplementary Data 1: Sequence data).

Additional points addressing the supplementary .docx file containing Supplementary Figures, Tables and Notes:

7) page 35 line 480: "proteins from *Arabidopsis thaliana* TAIR10"

Are you sure it is TAIR10 and not data from Araport11? Even if recently downloaded from TAIR, the content might be from Araport. TAIR did take over data from Araport11 after funding for Araport stopped.

8) page 36 line 541: "orthologous gene clusters among a broader set of monocots: ..."
The set also lists *A. thaliana* which is a dicot.

9) page 48 line 1017:

Reference 75 has some typos:

"Rabl, C. *Über Zellteilung. Morphologisches Jahrbuch* 214–330 (1885)."

It should be:

Rabl, C. *Über (Ueber) Zelltheilung, Morphologisches Jahrbuch* 214–330 (1885).

Reviewer #2:

Remarks to the Author:

In the manuscript "Chromosome evolution and the genetic basis of agronomically important traits in greater yam," Bredeson et al., describe a high-quality, chromosome-resolved genome for *Dioscorea alata* that they leverage to identify quantitative trait loci (QTL) for disease susceptibility and tuber quality as well as better understand the genomic architecture underlying the breeding history. First, the authors describe their high-quality genome and how they leveraged both HiC and genetic maps to accurately resolve the 20 chromosomes. Next the authors compare their genome to other recently published genomes of closely related yams, which provides support for the quality of their assembly, and identifies the delta whole genome duplication (WGD) in context of the monocot-specific tau WGD. The authors then go on to map QTL for Yam Anthracnose Disease (YAD) and tuber quality in multi-year and bi-parental populations, with the identification of potential genomic regions that could be used for molecular breeding; the authors do note that their regions do not overlap with other studies. Finally, the authors leverage resequencing of several lines to define regions of high and low heterozygosity, which provides insight into the breeding history and potential introgressions responsible for interesting phenotypes. This is an excellently crafted and written manuscript, which will greatly benefit both the yam breeding community and the broader plant biology community. Minimal suggestions are listed below.

The authors identify QTL and then areas of introgression. There is not a synthesis of these two sections. Do the QTL regions intersect with regions of introgression? Does this provide clues as to selection on the disease and tuber traits? Also, the authors perform a very nice K_s analysis that could be extended in the close relatives to identify proteins that are under selection. It would be very interesting to know if genes under selection are in the homozygous or heterozygous regions as a test of the significance of those regions. While there was a little discussion of the genes that underly the tuber quality QTL, there is not mention of the possible genes under the YAD QTL. Is this due to the fact there are just so many disease genes under those peaks? Once again this would be very interesting to harmonize with the genomic architecture part. I really appreciate that the authors have addressed the incongruences between their QTL studies and previously published QTL and GWAS

studies (and differences over the years of their study). Perhaps this could be augmented a bit so the reader may understand better why this may be heritability of traits etc since that data is available. There are many studies where QTLs are identified without reference to the genome (in general across plants), and when a high-quality genome is available (ie like in a genome focused paper like this one), there needs to be a tie back to the genome; otherwise, this is just another QTL study where a gene was not cloned or molecular markers were not demonstrated.

Figure 2d suggestion very interesting in terms of how young the diploidization is from the delta tetraploidy. How does the fractionation compare across the Dioscorea? If there are significant differences in fractionation, especially in the regions of differing heterozygosity, this could be very interesting for downstream breeding/QTL efforts and molecular analyses. Is there any connection between introgression/heterozygosity/QTL with the changes in chromosome structure, ie Chrs 5, 6, 18, 19? Likewise, the fractionation with *D. zingiberensis* that shares the delta tetraploidy but has already reduced to 10 chromosomes.

Minor edits

Line 110 "Rab1" should be defined here (or earlier); will help with the flow (for the reader).

Line 170 needs a period after "(see also refs)."

Lines 429-431 panel d could use a legend; it is in the text but would be easier to follow if also in the figure

Todd Michael

Reviewer #3:

Remarks to the Author:

Bredeson et al report a chromosome scale genome of the greater yam, *Dioscorea alata*, a species known to provide food and used around the world. The authors report a lineage specific WGD event shared by Dioscoreales. They also placed the previously known 'tau' WGD event by comparing synteny to other high quality monocot genomes. The authors also found multiple significant QTLs for disease resistance and ect. The paper is clear and well-written and I have only minor comments/concerns.

The quality of the genome assembly and annotation is very impressive. The authors did a nice job to make this genome available on phytozome and the yambase. This genome will be an important resource for people studying yams or interested in using yam genomes for other comparative studies.

The paleopolyploidy analyses are solid. I was very interested in how they infer the ancient polyploidy events and went into the two papers they cited in this paper (Ref 22 Cheng et al. 2021 and Ref 35 Ren et al. 2018) as support for previous evidence of ancient polyploidy. In fact, Ren et al. 2018 completely missed the ancient WGD in Dioscorea, as mentioned in Cheng et al. 2021. The citation here should be corrected. I also checked the 1KP consortium 2019 paper (One thousand plant transcriptomes and the phylogenomics of green plants), as they recently looked at the ancient polyploidy across many plant transcriptomes. The 1KP inferred the 'delta' event (DIV1a in the 1kp paper) from the transcriptome of *Dioscorea villosa* (which is also missed by Cheng et al. 2021). The placement of the 'tau' event (ORSAγ in the 1kp paper) is also consistent with between the 1kp and Bredeson et al. It is very nice to see these two papers using completely different data and wgd analyses but found the same results. I suggest the authors correct the references and add a sentence for comparing to the 1kp results.

In page 10 line 271, The author wrote 'Dioscoreales is one of the earliest branching core monocots'. This is inaccurate in a phylogenetic sense, as looking at SI fig.9 Dioscoreales is branching together with other major core monocot lineages. I suggest rewording the sentence for accuracy.

Reviewer #1:

The manuscript by Bredeson et al. entitled "Chromosome evolution and the genetic basis of agronomically important traits in greater yam" presents a comprehensive and high-quality study centred on the genome sequence of *Dioscorea alata* (greater yam). The genome sequence has been generated based on SMRT long reads (PacBio), has reached chromosome arm quality including quite some telomeres (centromeres missing), and has been ordered based on a newly generated high resolution genetic map. In general, a very convincing study!

The analyses regarding chromosome evolution in the *Dioscorea* species yielded interesting results regarding a "delta" and a "tau" WGD, these are well documented in the results (and the comprehensive supplements) and are also nicely put into literature context.

The QTL analyses are substantial and have been performed with quite some resources, however, the results are not that impressive. Although there is one candidate QTL locus on chromosome 5 (see Fig. 3) explaining "48.2% of phenotypic variance in the 2017 data", this QTL was not validated in other years or other population. Given the problems which are correctly mentioned and very appropriately discussed in the paper, the statement in the Concluding Remarks regarding "facilitat[ing] marker-assisted breeding in this crop [*D. alata*]" seems a bit far fetched. Nevertheless, it is very obvious that the genomic and genetic resources provided by this paper will greatly support *D. alata* breeding.

The results and data presented are novel and are surely relevant for a larger audience.

Thank you for your careful and thoughtful reading of our manuscript and supplementary materials! We are very pleased with the quality of the *D. alata* genome and our findings. We have added a new analysis of differential gene loss that shows that the *Dioscorea*-specific "delta" duplication was an allotetraploidy. Regarding centromeres, we now note that we identify unique centromeric tandem repeat loci within all chromosomes except chromosome 1. These coincide with centromere positions estimated based on the Rab1 conformation. Chromosome 1 contains several centromeric tandem repeat arrays that are linked to, but cannot be precisely placed within, the highly-repetitive pericentromere. These findings are further support for the completeness of our genome assembly.

The concluding remark about "facilitat[ing] marker-assisted breeding" in *D. alata* was intended as a forward-looking statement about the future usefulness of the genome sequence, genetic markers, and map—we did not mean to imply that this would be limited to our QTL (which, as noted, may not replicate in other crosses). We have edited this to "We demonstrated the utility of these resources by finding eight QTL for anthracnose disease resistance and tuber quality traits in *D. alata*. The genome sequence and associated genetic resources will facilitate future marker-assisted breeding in this crop."

Specific points:

1) Throughout: The authors should clearly distinguish between the terms "genome" and "genome sequence". Only a genome sequence can be assembled and annotated. Only the

genome (as DNA in the cell) has the ability to become realised in a phenotype or trait. Reading the text is difficult when this distinction is blurred. The problem becomes very obvious in the paragraph "Comparative analysis and paleopolyploidy", but it starts already in the abstract.

Another point also addressing terminology and the use of designations. Throughout the text, it should be either "D. alata" OR "greater yam", but not a mix. Jumping between the two terms is confusing.

Thank you for this comment. As requested, we have now clearly distinguished between genome and genome sequence, as appropriate. Regarding usage of "D. alata" vs. "greater yam," we have standardized the main manuscript text to refer to "D. alata" when used in a phylogenetic and scientific context and reserve "greater yam" for use in a cultural context. We hope this will not create any confusion, as we state in the abstract and introduction that the binomial and common name refers to the same species.

2) It must be made more obvious to the reader which published sequences were used for comparisons. This information is presented in "Supplementary Table 7" and is mentioned here and there in the supplements, but the references and identifiers must be presented clear and unambiguous also in the main text (e.g. page 7 line 178). It is not sufficient to add more pointers to "Supplementary Table 7".

Thank you for this comment. The publications reporting these other sequences were cited in the introduction (page 3, line 58) but we should of course cite them again in the main body of the manuscript text to make clear that they are the source of sequences used for comparative analyses. We have now added citations on page 8 (lines 193–194) and several other places in the text. We have now also included citations to these publications, and GenBank/RefSeq numbers in the Methods section, for each genome sequence that was used.

3) page 4 line 90: Citation of "Gatarira et al., in preparation)". It is not clear to this reviewer why this citation is included here. Supplementary Fig. 2 shows the chromosome count ($2n = 40$) for TDa95/00328, the sequenced reference genotype. The Supplementary Methods provide the respective protocol as required. If this citation is relevant, at least a preprint (at bioRxiv) needs to be provided and cited.

As the Reviewer notes, the chromosome count is supported by Supplementary Fig. 2 and the corresponding methods, so no "in preparation" citation is needed. We have, therefore, removed the citation to "Gatariria et al., in preparation", which will provide additional supporting flow cytometry data.

4) page 5 line 99: The text regarding the populations used to generate the genetic maps is difficult to follow. It should be made clear why there are different numbers mentioned at different places in the text:

- line 112: "ten genetic maps were highly concordant"

- Legend to Figure 1b: composite genetic linkage map (black points), integrating five mapping populations

- Table 2: 9 parent pairs listed, but 11 population IDs, and TDa1402 & TDa1506 & TDa1621 as well as TDa1419 & TDa1610 share identical parents

This reviewer assumes that rephrasing the text should clarify the questions marks.

Thank you for these comments. We are sorry for the confusion and have tried to further clarify the manuscript and supplementary text. For convenience, the accounting is summarized here:

As shown in Table 2, and described in more detail in Supplementary Note 2, there were 6 *crossing combinations* (i.e., distinct pairs of parental genotypes), from 7 distinct parental lines. Following convention, we distinguish 11 *populations* (denoted TDa####, a notation consistently used in Yambase and other databases containing yam breeding records) that each represent a distinct group of progeny generated from a pair of parental lines (which may have been crossed at different times) and/or are raised in different locations (e.g., IITA or NRCRI). So, in particular, two or more populations can be derived from the same parental pair.

We constructed ten F₁ genetic maps. Nine of these are for single populations; the tenth is a map combining two small populations (TDa1512 and TDa1603), since they (1) had the same parents and (2) TDa1603 had too few progeny (n=10) to make a separate map.

To make a framework map, we merged five F₁ maps that capture the genetic diversity of the seven distinct parents. We could not merge all ten maps because the algorithm for map combination, LPmerge, did not complete with larger datasets (based on the wall-clock limits of our compute server), as is now noted in Supplementary Note 2.

5) page 8 line 214: What is the evidence for n=20 for *D. dumetorum*? Neither data nor a reference is provided. It might be that the old estimation of n=18/2n=36 (Miege, Revue de Cytologie et de Biologie Végétales, 1954) is not correct. But if so, the authors need to present evidence for the correction.

Thank you for this comment. A-recent chromosome count for *D. dumetorum* (2n=40), listed in the Chromosome Counts DataBase (CCDB) and, is from Baquar, S. R. 1980. Chromosome behaviour in Nigerian yams (*Dioscorea*). *Genetica* 54: 1–9. A count of 2n=40 is also reported in Lebot, V. (2019). *Tropical Root and Tuber Crops, 2nd Edition*. CABI. We have now included these citations in the manuscript.

6) The sequence data for TDa95-310 and TDa95/00328 must be submitted as well (see Supplementary Data 1: Sequence data).

All WGS data generated for this work, including for TDa95-310 and TDa95/00328, are listed with their SRA accession numbers in Supplementary Data 1. The data are viewable at NCBI via the Reviewer Link provided in the Data Availability section of the manuscript. We have asked Saski et al. to deposit the data from their publication and understand that this is in progress.

Additional points addressing the supplementary .docx file containing Supplementary Figures, Tables and Notes:

7) page 35 line 480: "proteins from *Arabidopsis thaliana* TAIR10"
Are you sure it is TAIR10 and not data from Araport11? Even if recently downloaded from TAIR, the content might be from Araport. TAIR did take over data from Araport11 after funding for Araport stopped.

Thank you for this comment. Throughout, we have updated the *Arabidopsis* genome citation to the Arabidopsis Genome Initiative (2000) Nature paper (doi:10.1038/35048692) and the TAIR10 annotation cited as Lamesch, et al. (2012) *Nucleic Acids Research* (doi:10.1093/nar/gkr1090).

8) page 36 line 541: "orthologous gene clusters among a broader set of monocots: ..."
The set also lists *A. thaliana* which is a dicot.

Thank you for catching this. We have corrected this sentence, which mistakenly implied *A. thaliana* was a monocot.

9) page 48 line 1017:
Reference 75 has some typos:
"Rabl, C. Über Zellteilung. Morphologisches Jahrbuch 214–330 (1885)."
It should be:
Rabl, C. Über (Ueber) Zelltheilung, Morphologisches Jahrbuch 214–330 (1885).

Thank you! We have corrected this citation.

Reviewer #2:

In the manuscript “Chromosome evolution and the genetic basis of agronomically important traits in greater yam,” Bredeson et al., describe a high-quality, chromosome-resolved genome for *Dioscorea alata* that they leverage to identify quantitative trait loci (QTL) for disease susceptibility and tuber quality as well as better understand the genomic architecture underlying the breeding history. First, the authors describe their high-quality genome and how they leveraged both HiC and genetic maps to accurately resolve the 20 chromosomes. Next the authors compare their genome to other recently published genomes of closely related yams, which provides support for the quality of their assembly, and identifies the delta whole genome duplication (WGD) in context of the monocot-specific tau WGD. The authors then go on to map QTL for Yam Anthracnose Disease (YAD) and tuber quality in multi-year and bi-parental populations, with the identification of potential genomic regions that could be used for molecular breeding; the authors do note that their regions do not overlap with other studies. Finally, the authors leverage resequencing of several lines to define regions of high and low heterozygosity, which provides insight into the breeding history and potential introgressions responsible for interesting phenotypes. This is an excellently crafted and written manuscript, which will greatly benefit both the yam breeding community and the broader plant biology community. Minimal suggestions are listed below.

Thank you for your careful and thoughtful reading of our manuscript, and helpful comments addressed below!

The authors identify QTL and then areas of introgression. There is not a synthesis of these two sections. Do the QTL regions intersect with regions of introgression? Does this provide clues as to selection on the disease and tuber traits? Also, the authors perform a very nice Ks analysis that could be extended in the close relatives to identify proteins that are under selection. It would be very interesting to know if genes under selection are in the homozygous or heterozygous regions as a test of the significance of those regions. While there was a little discussion of the genes that underly the tuber quality QTL, there is no mention of the possible genes under the YAD QTL. Is this due to the fact there are just so many disease genes under those peaks? Once again this would be very interesting to harmonize with the genomic architecture part. I really appreciate that the authors have addressed the incongruences between their QTL studies and previously published QTL and GWAS studies (and differences over the years of their study). Perhaps this could be augmented a bit so the reader may understand better why this may be -- heritability of traits etc since that data is available. There are many studies where QTLs are identified without reference to the genome (in general across plants), and when a high-quality genome is available (ie like in a genome focused paper like this one), there needs to be a tie back to the genome; otherwise, this is just another QTL study where a gene was not cloned or molecular markers were not demonstrated.

Thank you for the many interesting suggestions! We checked the 90% LD region around each QTL for overlap with the putatively introgressed regions in the mapping parents but found no overlaps. This negative result is now mentioned briefly at the very end of Anthracnose Resistance subsection of the QTL Mapping section (page 15, lines 356–359).

We looked more closely at the genes in the anthracnose disease resistance QTL, and now include a brief discussion of several plausible LRR-containing genes and EMSY-like immune regulator genes, but a full investigation is beyond the scope of this paper. Resistance to anthracnose disease, its genes, and mechanisms have been most intensively studied in common bean and sorghum but remains poorly understood, and there have been no molecular studies in yam, so it is difficult to go further, but our genomic resources will enable such studies going forward. The full lists of genes, complete with functional annotations, underlying our QTL peaks are now provided in Supplementary Data 5.

Regarding the lack of congruence with other QTL studies, we note that these are generally dependent on the segregation of relevant genes in the map cross. Our map crosses are from ongoing field trials with parents that are already the result of improvement/inbreeding, they were not selected to highlight segregating YAD genes, as, for example, an F₂ cross from resistant and susceptible inbred varieties would do. We are not aware of any measurements of heritability of susceptibility of YAD in yam, which would in any event be dependent on the population that was studied (again highlighting the importance of the presence of segregating variants of relevant genes).

Figure 2d suggestion very interesting in terms of how young the diploidization is from the delta tetraploidy. How does the fractionation compare across the Dioscorea? If there are significant differences in fractionation, especially in the regions of differing heterozygosity, this could be very interesting for downstream breeding/QTL efforts and molecular analyses. Is there any connection between introgression/heterozygosity/QTL with the changes in chromosome structure, ie Chrs 5, 6, 18, 19? Likewise, the fractionation with *D. zingiberensis* that shares the delta tetraploidy but has already reduced to 10 chromosomes.

Thank you for this comment, which prompted us to do additional analysis of differential gene loss between homoeologous chromosomes to try to infer whether *D. alata* and other *Dioscorea* species are paleo-autotetraploids or paleo-allotetraploids. Our analyses, based on a simple (and to our knowledge, new) test for consistently asymmetric gene loss across homeologs, suggests that the Dioscoreaceae-specific genome duplication was an ancient allotetraploidy event. This is now described in the manuscript (pages 11–12, lines 262–284) related to the Dioscoreaceae-specific genome duplication.

Briefly, we estimated differential gene loss after genome duplication (sometimes referred to as “fractionation”) by considering 15 pairs of nearly collinear homoeologous segments with more than 40 total homoeologous gene pairs. These segments are found on 11 distinct homoeologous chromosome pairs (new Supplementary Table 6). We considered genes with homoeologous paralogs to be retained after genome duplication. Single copy genes were considered to have lost their homoeologous paralog after whole-genome duplication. In the absence of a suitable unduplicated outgroup (all related monocot lineages have their own complicated duplication history), we considered only genes with orthologs in five or more other monocots to avoid counting spurious gene models as single-copy

retained genes. We used these methods to estimate the retention rates of these 15 pairs of homoeologous segments.

We found that (1) the distribution of retention rates across the resulting 30 chromosomal segments is bimodal with peaks at 0.48 and 0.63 (Supplementary Fig. 9), and (2) after collapsing these 15 segments into 11 distinct homoeologous chromosome pairs, each pair has one chromosome from the “high” mode and one from the “low” mode (new Supplementary Table 6). (“High” and “low” were determined relative to 0.55, the median of the 22 retention rates). Under the null model in which the retention rate of each segment is randomly chosen from the bimodal distribution, regardless of the rate of its homoeolog, the probability that each of the 11 pairs consist of high and low retention segments is $2^{11} / \binom{22}{11} \approx 3 \times 10^{-3}$. This non-parametric analysis allows us to reject the null hypothesis in favor of a “biased fractionation” model, which implies ancient allotetraploidy. (A more complicated parametric analysis similarly supports this model of biased gene loss.)

Repeating this analysis for other *Dioscorea* genomes yields consistent results (new Supplementary Table 7, Supplementary Note 5). The fact that the great majority of genes have orthologs across the *Dioscorea* species (Supplementary Fig. 6) suggests that most gene loss after the Dioscoreaceae genome duplication occurred relatively rapidly and prior to the radiation of extant *Dioscorea* species. We did not attempt to identify lineage-specific losses within sequenced *Dioscorea* spp. genomes, since this would be easily confounded by challenges of gene annotation, differing assembly qualities, etc.

Minor edits

Line 110 “Rab1” should be defined here (or earlier); will help with the flow (for the reader).

Thank you, we have added a brief definition of the Rab1 conformation at its first mention (lines 145–147).

Line 170 needs a period after “(see also refs).”

Thank you!

Lines 429-431 panel d could use a legend; it is in the text but would be easier to follow if also in the figure

Thank you for this comment. The legend for Figure 4d, with dark blue, cyan, and orange representing IBD0, IBD1, and IBD2 (respectively), has been moved to a more prominent position.

Todd Michael

Reviewer #3 (Remarks to the Author):

Bredeson et al report a chromosome scale genome of the greater yam, *Dioscorea alata*, a species known to provide food and used around the world. The authors report a lineage specific WGD event shared by Dioscoreales. They also placed the previously known 'tau' WGD event by comparing synteny to other high quality monocot genomes. The authors also found multiple significant QTLs for disease resistance and ect. The paper is clear and well-written and I have only minor comments/concerns.

Thank you!

The quality of the genome assembly and annotation is very impressive. The authors did a nice job to make this genome available on phytozome and the yambase. This genome will be an important resource for people studying yams or interested in using yam genomes for other comparative studies.

Thank you.

The paleopolyploidy analyses are solid. I was very interested in how they infer the ancient polyploidy events and went into the two papers they cited in this paper (Ref 22 Cheng et al. 2021 and Ref 35 Ren et al. 2018) as support for previous evidence of ancient polyploidy. In fact, Ren et al. 2018 completely missed the ancient WGD in *Dioscorea*, as mentioned in Cheng et al. 2021. The citation here should be corrected. I also checked the 1KP consortium 2019 paper (One thousand plant transcriptomes and the phylogenomics of green plants), as they recently looked at the ancient polyploidy across many plant transcriptomes. The 1KP inferred the 'delta' event (DIVI α in the 1kp paper) from the transcriptome of *Dioscorea villosa* (which is also missed by Cheng et al. 2021). The placement of the 'tau' event (ORSA in the 1kp paper) is also consistent with between the 1kp and Bredeson et al. It is very nice to see these two papers using completely different data and wgd analyses but found the same results. I suggest the authors correct the references and add a sentence for comparing to the 1kp results.

Thank you for this careful discussion and pointer to the "1KP" analyses of WGDs. This extensive transcriptome-based study is complementary to our inferences based on chromosomal paralogy. We have revised our discussion of duplications to reflect these points noting the correspondence of 'delta' with DIVI-alpha and 'tau' with ORSA-gamma of the 1KP consortium (lines 236–241).

In page 10 line 271, The author wrote 'Dioscoreales is one of the earliest branching core monocots'. This is inaccurate in a phylogenetic sense, as looking at SI fig.9 Dioscoreales is branching together with other major core monocot lineages. I suggest rewording the sentence for accuracy.

Thank you for catching this. We have revised the sentence to make it clear that Dioscoreales are one of several major core monocot lineages that diverged more or less at the same time (to the resolution of current phylogenies).

Reviewers' Comments:

Reviewer #1:

Remarks to the Author:

The revised version has been significantly improved, all my points and remarks are fully covered. No additional comments.

Reviewer #2:

Remarks to the Author:

The authors have addressed my comments and more. It looks great.

Reviewer #3:

Remarks to the Author:

The author addressed my comments.

However, their revision went a little bit too far for concluding the 'delta' WGD is an ancient allopolyploid. I agree that they found some interesting pattern of subgenome dominance. And it has been hypothesized that genomes with evidence of biased fractionation and subgenome dominance are more likely to be ancient allopolyploids (Garsmeur et al. 2014). This is an elegant hypothesis, however, recent studies have shown that this pattern is not universal. Multiple well recolonized allo species (e.g. Brassica napus, wheat, and cotton) does not shown biased fractionation and subgenome dominance. The parental genome divergence, differences in life history, TE density and methylation pattern, and etc. can also be responsible for subgenome dominance, but it does not require the allopolyploidy nature of the WGD.

Reviewer #3 writes:

The author addressed my comments.

However, their revision went a little bit too far for concluding the 'delta' WGD is an ancient allopolyploid. I agree that they found some interesting pattern of subgenome dominance. And it has been hypothesized that genomes with evidence of biased fractionation and subgenome dominance are more likely to be ancient allopolyploids (Garsmeur et al. 2014). This is an elegant hypothesis, however, recent studies have shown that this pattern is not universal. Multiple well recolonized [recognized?] allo species (e.g. *Brassica napus*, wheat, and cotton) does not shown biased fractionation and subgenome dominance. The parental genome divergence, differences in life history, TE density and methylation pattern, and etc. can also be responsible for subgenome dominance, but it does not require the allopolyploidy nature of the WGD.

Thanks for this comment! We appreciate this point and have modified the text to make our assumptions clear and to acknowledge the possible limitations of our methodology.

In the analysis noted above by Reviewer #3 (added in response to an earlier comment by Reviewer #2), we found that the *Dioscorea* genome can be partitioned into two homoeologous chromosome sets (subgenomes) such that one set consistently retains more ancestral genes than the other, i.e., shows consistent “fractionation bias.” Such a bias is not expected under an autotetraploid model, for which homoeologous chromosomes are interchangeable, that is, have the same TE density, methylation pattern, etc. immediately after genome doubling, and are therefore not expected to evolve asymmetrically after polyploidy. Specifically, our statistical test rejects the hypothesis of unbiased gene loss that would be expected under an autotetraploid model. Since we can reject autotetraploidy, the simplest explanation for this observation is that yam is a paleo-allotetraploid.

Consistently biased gene loss is a *positive* signal of paleo-allotetraploidy because hybridization of two distinct progenitors establishes an initial asymmetry between subgenomes that *may* become amplified over time by subsequent systematic patterns of differential gene loss and expression change, as first noted by Garsmeur et al.¹. We emphasize “may” because the initial asymmetry that arises from interspecific hybridization is only *permissive* of the (still somewhat mysterious) processes that amplify initial differences between subgenomes over time. For example, if the two progenitors were very similar, or if not enough time had elapsed since allotetraploidization, significant differences between subgenomes may not be detectable.

Thus while subgenome bias is positive evidence for paleopolyploidy (as we find for *Dioscorea*), the absence of detectable sub-genome bias could be consistent with either an auto- or allo-polyploidization mechanism, and we have attempted to make this point more clearly.

As noted by Reviewer #3, there are several polyploids that are believed to have arisen by an allopolyploid mechanism but that do not show biased gene loss. This is an area of active ongoing research and rather than provide a detailed discussion of this point we have added citations to two review articles (Cheng et al.² and Wendel et al.³) to alert the reader to ongoing work and the potential for allotetraploidy without biased fractionation.

Regarding the examples noted by Reviewer #3, wheat, cotton, and *B. napus* were all formed by “recent” interspecific hybridization (several tens of thousands of years or less) and have their own special features. A full discussion of these cases is beyond the scope of our *Dioscorea* paper (and this response!), and we have accordingly cited review articles in the revised text, but make some notes below that appear to be relevant to the Reviewer’s point.

- Hexaploid bread wheat appears to have been formed by hybridization of an AABB tetraploid (emmer wheat) and a DD genome that itself is derived by homoploid hybridization of AA and BB-like progenitors⁴, so that hexaploid wheat behaves in some sense like a segmental mixture of AAAABB and AABBBB segments, which is outside of the scope of our simple model. The original IWGSC⁵ wheat genome paper notes that the three subgenomes have comparable gene content, suggesting no bias in gene loss; and more recently, Jeury et al.⁶ “hypothesize that, unlike most of the allopolyploid species, subgenome dominance and biased fractionation are absent in hexaploid wheat” but note that this point is controversial, citing prior work by Pont and Salse⁷ that described “asymmetrical genome evolution” in bread wheat. In any event, as a recent polyploid, gene losses in hexaploid wheat are expected to be small (Jeury et al. estimate 450 per genome at each step of polyploidization, with A and B experiencing two steps and therefore more loss, relative to one step in D). These small losses are consistent with the very recent formation of the hexaploid and make statistically significant differential loss difficult to discern.
- Similarly, in *B. napus*, Chalhoub et al.⁸ noted that the vast majority of genes are found in both the A- and C-subgenomes, observing “the loss of 112 An and 91 Cn genes in *B. napus* ‘Darmor-bzh’”. With such a low level of gene loss, the asymmetry would have to be very large to be considered significant. This is as might be expected for such a polyploid that formed only 7,500 years ago.
- Finally, *Gossypium* spp. (cotton) is cited by Wendel et al.³ as an example of a paleotetraploid that *does* show biased fractionation (see also Renny-Byfield et al.⁹). Since cotton has multiple tetraploid forms, some of them may not show sub-genome biased gene loss.

References

1. Garsmeur, O. *et al.* Two evolutionarily distinct classes of paleopolyploidy. *Mol. Biol. Evol.* **31**, 448–454 (2014).
2. Cheng, F. *et al.* Gene retention, fractionation and subgenome differences in polyploid plants.

Nat Plants **4**, 258–268 (2018).

3. Wendel, J. F., Lisch, D., Hu, G. & Mason, A. S. The long and short of doubling down: polyploidy, epigenetics, and the temporal dynamics of genome fractionation. *Curr. Opin. Genet. Dev.* **49**, 1–7 (2018).
4. Marcussen, T. *et al.* Ancient hybridizations among the ancestral genomes of bread wheat. *Science* **345**, 1250092 (2014).
5. International Wheat Genome Sequencing Consortium (IWGSC). A chromosome-based draft sequence of the hexaploid bread wheat (*Triticum aestivum*) genome. *Science* **345**, 1251788 (2014).
6. Juery, C. *et al.* New insights into homoeologous copy number variations in the hexaploid wheat genome. *Plant Genome* **14**, e20069 (2021).
7. Pont, C. & Salse, J. Wheat paleohistory created asymmetrical genomic evolution. *Curr. Opin. Plant Biol.* **36**, 29–37 (2017).
8. Chalhoub, B. *et al.* Plant genetics. Early allopolyploid evolution in the post-Neolithic *Brassica napus* oilseed genome. *Science* **345**, 950–953 (2014).
9. Renny-Byfield, S., Gong, L., Gallagher, J. P. & Wendel, J. F. Persistence of subgenomes in paleopolyploid cotton after 60 my of evolution. *Mol. Biol. Evol.* **32**, 1063–1071 (2015).